# Analytical solution of nitracline with the evolution of subsurface chlorophyll maximum in stratified water columns

Xiang Gong[1, 2], Wensheng Jiang[2, 3], Linhui Wang[2], Huiwang Gao[2, 4]*, Emmanuel Boss[5], Xiaohong Yao[2,4], Shuh-Ji Kao[6], Jie Shi[2]

[1] School of Mathematics and Physics, Qingdao University of Science and Technology, Qingdao 266061, P. R. China

[2] Key Laboratory of Marine Environment and Ecology (Ministry of Education of China), Ocean University of China, Qingdao 266100, P. R. China

[3] Key Laboratory of Physical Oceanography (Ministry of Education of China), Ocean

University of China, Qingdao 266100, P. R. China

[4] Qingdao Collaborative Center of Marine Science and Technology, Ocean University of China, Qingdao 266100, P. R. China

[5] School of Marine Sciences, University of Maine, Orono 04469-5706, USA

[6] State Key Laboratory of Marine Environmental Science, Xiamen University, Xiamen

361005, P. R. China

*Corresponding author: Huiwang Gao, hwgao@ouc.edu.cn

**Abstract:**

In a stratified water column, the nitracline is a layer where the nitrate concentration

increases below the nutrient-depleted upper layer, exhibiting a strong vertical gradient in the euphotic zone. The subsurface chlorophyll maximum layer (SCML) forms near the bottom of euphotic zone, acting as a trap to diminish the upward nutrient supply. Depth and steepness of the nitracline are important measurable parameters related to the vertical transport of nitrate into the euphotic zone. The correlation between the

SCML and the nitracline has been widely reported in the literature, but the analytic solution for the relationship between them is not well established. By incorporating a piecewise function for the approximate Gaussian vertical profile of chlorophyll, we derive analytical solutions of a specified nutrient-phytoplankton model. The model is well suited to explain basic dependencies between a nitracline and a SCML. The

analytical solution shows that the nitracline depth is deeper than the depth of SCML, shoaling with an increase in light attenuation coefficient and with a decrease in surface light intensity. The inverse proportional relationship between the light level at the nitracline depth and the maximum rate of new primary production is derived. Analytic solutions also show that a thinner SCML corresponds to a steeper nitracline.

The nitracline steepness is positively related to light attenuation coefficient, but independent of surface light intensity. The derived equations of the nitracline in relation to the SCML provide further insight into the important role of the nitracline in marine pelagic ecosystems.

## 1 Introduction

Nitrogen availability, especially the nitrate upward supply to the euphotic zone where light intensity is sufficient to support net photosynthesis, limits the primary productivity in a stratified water column (Falkowski et al., 1998). Specifically, the nitrate supply from below and the light attenuated from above with the depth collaboratively affect the growth of phytoplankton and lead to the subsurface chlorophyll maximum (SCM) (Riley et al., 1949; Steele and Yentsch, 1960; Herbland and Voituriez, 1979; Cullen, 1982). The SCM layer (SCML) has attracted much attention since Riley (1949) because the layer contributes significantly to new primary production (NPP) in stratified waters (Probyn et al., 1995; Ross and Sharples, 2007; Fernand et al., 2013). The synergistic physical and biological interaction leads to a strong vertical nitrate gradient, conventionally referred to as the nitracline (Eppley et al., 1978; Herbland and Voituriez, 1979; Cullen and Eppley, 1981). Depth and steepness of the nitracline are important measurable parameters in regulating the supply of nitrate to the euphotic zone, and hence affecting NPP (Lewis et al., 1986; Bahamón et al., 2003; Aksnes et al., 2007; Cermeno et al., 2008; Omand and Mahadevan, 2015).

The nitracline depth physically depends on the degree of water-column stratification and the magnitude of momentum transfer associated with wind stress (Denman and Gargett, 1983; Laanemets et al., 2004). It also depends on momentum transfer from below (Lipschultz et al., 2002) and in some cases, vertical advection such as upwelling (Laanemets et al., 2004). However, in a relatively stable environment, the SCML may restrict the diffusive flux of nitrates to the euphotic zone and continually erode the nitracline supposing that sufficient light is available (Probyn et al., 1995). The SCML thereby acts as an effective nutrient trap, regulating the nitracline depth (Banse, 1982; Beckmann and Hense, 2007; Klausmeier and Litchman, 2001; Probyn et al., 1995). However, variation of nitracline steepness, which is critical to determine the nitrate supply, was poorly understood due to lack of high vertical resolution data, e.g., both bottle data and Argo data tend to have low vertical

resolution sampling. Some studies showed that nitrogen flux is dependent more on the nitracline steepness than on the density gradients regulating turbulent diffusion (Bahamón and Cruzado, 2003; Bahamón et al., 2003; Lavigne et al., 2015). Thus, these measurable features of nitracline and their correlation with SCML may provide insightful information for mechanisms of the productivity in pelagic ecosystem, and the analytic solutions for these parameters may fill the knowledge gap.

Although a close relationship between the nitracline and SCML is always observed, the quantitative nature of nitracline in relation to the SCML formation has not been studied. The system of phytoplankton and the limiting nutrient on the vertical axis was often utilized to study the depth, intensity, and persistence of the SCML. Major theoretical results include photoacclimation (increase of chlorophyll per cell) (Steele, 1964; Fennel and Boss, 2003), bistability (Yoshiyama and Nakajima, 2002; Ryabov et al., 2010), oscillating SCM (Huisman et al., 2006), hysteresis conditions (Kiefer and Kremer, 1981; Navarro and Ruiz, 2013), and the ESS (evolutionary stable strategy) depth obtained by game-theory approach (Klausmeier and Litchman, 2001; Mellard et al., 2011). Recent mathematical studies solved the persistence and uniqueness of the steady state solution (Du and Hsu, 2010; Hsu and Yuan, 2010; Du and Mei, 2011), and gave rigorous proofs for the above-mentioned ESS depth and the game-theory approach (Du and Hsu, 2008a, b). Additionally, several modeling studies have been conducted to quantitatively assess the importance of different physical-biological processes leading to SCML (Jamart et al., 1977; Jamart et al., 1979; Varela et al., 1994; Klausmeier and Litchman, 2001; Hodges and Rudnick, 2004; Beckmann and Hense, 2007).

Among the studies using the nutrient-phytoplankton model, Klausmeier and Litchman (2001) first analytically derived the vertical nutrient distribution with the development of the SCML. In that model, the concentration of the limiting nutrient was found to be low and constant above the SCML and linearly increasing with depth below this layer in poorly mixed water column. Building on that model, Mellard et al. (2011) added stratification and surface nutrient input, which can make phytoplankton

grow in both the surface mixed layer and deep layer (SCML) simultaneously. Fennel and Boss (2003) derived that the sum of nutrients and phytoplankton at steady state will increase monotonically below the surface mixed layer until it equals the fixed nutrient concentration. By incorporating a generalized Gaussian function for vertical chlorophyll profile into the nutrient-phytoplankton dynamic equation, Gong et al. (2015) obtained that the steady-state nitrate concentration increased from the upper community compensation depth to the SCML depth. None of the studies, however, focused on the quantitative nature of nitracline in relation to the SCML in the stratified waters.

In this paper, we modified the nutrient-phytoplankton model by Gong et al. (2015) to study the roles of SCM in reshaping the nitracline. Two additional terms, atmospheric input, which promotes the growth of phytoplankton in the surface mixed layer, and the phytoplankton self-shading, which regulates the light penetration, were introduced into the previous model. Accordingly, a piecewise function comprising a constant value within the surface mixed layer and a Gaussian function below this layer was used as a fit to the steady state vertical chlorophyll profiles simulated by the nutrient-phytoplankton model. By incorporating the piecewise function into the nutrient-phytoplankton model, we derived the analytic solutions for the properties of the nitracline and the SCML in steady state, and the relationship between them was examined in response to light availability, surface nutrient input, and vertical diffusivity.

## 2 Definitions and Models

### 2.1 Models

#### 2.1.1 Dynamic equations

We consider the following equations for phytoplankton and nutrient dynamics in stratified waters (Eqs. 1-2), where light and nitrogen are two limiting factors for phytoplankton growth (Fig. 1). The change in phytoplankton at depth $z$ is the balance of the growth and death, and the passive moving (sinking and mixing) (Eq. 1). An

eddy diffusion coefficient $K_v$ redistributes phytoplankton in the water column. Depth $z$ is increasing toward the seabed.

$$\begin{cases} \dfrac{\partial P}{\partial t} = \mu_m \, \mathrm{min}\left(f\left(I\right) \; g\left(N\right)\right)P - \varepsilon P - w\dfrac{\partial P}{\partial z} + \dfrac{\partial}{\partial z}\left(K_v\left(z\right)\dfrac{\partial P}{\partial z}\right) \\ \dfrac{\partial N}{\partial t} = -\gamma\mu_m \, \mathrm{min}\left(f\left(I\right) \; g\left(N\right)\right)P + \gamma\alpha\varepsilon P + N_{in}\left(z\right) + \dfrac{\partial}{\partial z}\left(K_v\left(z\right)\dfrac{\partial N}{\partial z}\right) \end{cases}$$

where $P$ denotes the chlorophyll concentration (mg m$^{-3}$). We assume the chlorophyll distribution to represent the distribution of phytoplankton biomass (that means that the photoacclimation of phytoplankton is ignored, and the SCM refers to the subsurface biomass maximum, SBM). This is a significant simplification. In fact, phytoplankton increases inter-cellular pigment concentration when light level decreases (Cullen, 1982; Fennel and Boss, 2003; Cullen, 2015). Usually, the depths of SCML and SBML are separate, and the latter is shallower than the former.

Nitrogen $N$ (in unit: mmol N m$^{-3}$) taken up by phytoplankton includes three sources, i.e., recycling from dead phytoplankton, atmospheric input to the surface mixed layer, and supply by mixing from deep water (Eq. 2). $\gamma$ is the nitrogen content of the phytoplankton (mmol N per mg Chl). Following Mellard et al. (2011), we consider the nitrogen input from atmosphere at the rate of $N_{in}(z)$, setting as a delta function at $z=0$ with the total nutrient input in the surface mixed layer $N_{inML}$, $N_{inML} = \int_0^{z_s} N_{in}(z)dz$, where $z_s$ is the depth of surface mixed layer. Note that nitrogen input through the activity of nitrogen fixers is excluded. However, Trichodesmium, if they are at the surface, will be modeled similarly to the atmospheric input term.

$\mu_m$ is the maximum growth rate of phytoplankton, $\varepsilon$ is the loss rate of phytoplankton (including respiration, mortality, zooplankton grazing), and $\alpha$ is the recycling efficiency of dead phytoplankton ($0<\alpha<1$). The specific rate of loss processes ($\varepsilon$) is assumed to be vertically homogeneous due to lack of data (similar to Sverdrup, 1953). The treatment of grazing loss, is, in the least, an oversimplification, though many numerical models used a similar one (e.g., Klausmeier and Litchman, 2001; Fennel and Boss, 2003; Huisman et al., 2006). The growth-limited function for light $I$ and

nitrogen $N$ is given by $\mu_m \min(f(I), g(N))$. The Michaelis-Menten form for the light-limiting function $f(I)$ and nitrogen-limiting function $g(N)$, $f(I)=I/(K_I+I)$ and $g(N)=N/(K_N+N)$ is used, where $K_I$ and $K_N$ denote the half-saturation constants for light and nitrogen, respectively. The net growth rate, $\mu_m \min\left(f(I), g(N)\right)-\varepsilon$, is positive only if both light-limiting term $\mu_m f(I)$ and nitrogen-limiting term $\mu_m g(N)$ are larger than the loss rate $\varepsilon$.

Light intensity $I$ decreases exponentially with depth according to the Lambert-Beer law,

$$I(z) = I_0 \exp\left(-K_w z - K_c \int_0^z P dz\right) \tag{3}$$

where $I_0$ is the surface light intensity, $K_w$ and $K_c$ are light attenuation coefficients of water and phytoplankton, respectively. For the sake of simplicity, we assume that both $K_w$ and $K_c$ are constant with depth.

The sinking velocity of phytoplankton $w$ is non-negative in the chosen coordinate system. We assumed it to be constant with depth, neglecting the influencing of density gradients (pycnoclines), which may cause vertical variations in sinking.

To describe the water-column stratification, we assume that the vertical eddy diffusivity $K_v$ depends on depth,

$$K_v(z) = \begin{cases} K_{v1} & 0 \le z \le z_s \\ K_{v2} & z_s < z < z_b \end{cases} \tag{4}$$

where $z_b$ is set to the bottom boundary of this model and is assumed to be sufficiently deep where the chlorophyll concentration approaches to zero. We assume that surface diffusivity ($K_{v1}$) and subsurface diffusivity ($K_{v2}$) (Lande and Wood, 1987; Hodges and Rudnick, 2004) are constant and $K_{v1}$ is large enough to homogenize the chlorophyll and nitrogen concentration in the surface mixed layer. A gradual transition from the surface mixed layer to the deep one (in the vicinity of $z_s$) can be written in terms of a generalized Fermi function (Ryabov et al., 2010), $K_v(z) = K_{v2} + \frac{K_{v1}-K_{v2}}{1+e^{z-z_s/l}}$, where parameter $l$ characterizes the width of the transition layer. In our numerical study, we

assumed this transition layer is 2 m. For simplicity, parameter $l$ is set to infinitely thin in analytic solutions. A comprehensive list of symbols is given in Table 1.

The zero-flux boundary condition for the phytoplankton at the surface is used. At the bottom boundary of the model domain ($z=z_b$) the Dirichlet or fixed concentration boundary condition is used, i.e., $P \rightarrow 0$ for $z \rightarrow z_b$. Fennel and Boss (2003) used an infinite depth as $z_b$ ($z_b \rightarrow \infty$). For the nitrogen distribution we set $N=N_{inML}$ at the surface and diffusing into the water column a prescribed nitrate gradient, $n$, at the bottom. That is,

$$\begin{cases} \left( wP - K_{v1} \dfrac{\partial P}{\partial z} \right)\Big|_{z=0} = 0, \quad P(z_b) = 0. \\ N\big|_{z=0} = N_{inML}, \qquad\qquad \dfrac{\partial N}{\partial z}\Big|_{z=z_b} = n. \end{cases} \tag{5}$$

### 2.1.2 Fitting equation for vertical chlorophyll profile

In many stratified water columns, the vertical distribution of chlorophyll concentration is homogeneous within the surface mixed layer and appears a Gaussian below this layer (Fig. 2a), which is typical in open oceans (Uitz et al., 2006), shelf seas (Sharples et al., 2001), stratified estuary (Lund-Hansen, 2011), and arctic waters (Martin et al., 2012). The non-uniform vertical profile of chlorophyll within an SCML was first modeled by a generalized Gaussian function (Lewis et al., 1983), which has subsequently been widely used with small modifications. For example, Platt et al. (1988) superimposed a constant background on the generalized Gaussian, and fitted it to field data on the vertical distribution of chlorophyll from coastal, upwelling, open oceans and Arctic waters. Afterward, some studies introduced a parameter to represent the slope of the Gaussian curve (Matsumura and Shiomoto, 1993; Mu Oz Anderson et al., 2015). In particular, to account for the observed characteristic that surface values always exceed the bottom ones (Fig. 2a), the generalized Gaussian functional form has been modified with a superimposition of a background linearly or exponentially decreasing with depth (Uitz et al., 2006; Mignot et al., 2011; Ardyna et al., 2013).

For simplicity, to analytically study the role of the SCML on shaping the nitracline,

we therefore propose a piecewise function comprising a constant value in the surface mixed layer and below a general Gaussian function (Eq. 6) to approximate the vertical profile of chlorophyll concentration in Fig. 2a.

$$P = \begin{cases} P_0 & 0 \le z \le z_s \\ P_{max} \exp\left[ -\dfrac{\left(z - z_m\right)^2}{2\sigma^2} \right] & z_s < z < z_b \end{cases} \tag{6}$$

where $P$ is chlorophyll concentration as a function of depth $z$, $P_0$ is the chlorophyll concentration within the surface mixed layer, and $P_{max} = h/\left(\sigma\sqrt{2\pi}\right)$ represents the maximum value of chlorophyll below the surface mixed layer. Considering the influence of the surface mixed layer on the chlorophyll vertical distribution, $h$ is less than the total chlorophyll concentration integrated through the water column. Note that the vertical distribution of chlorophyll is an incomplete general Gaussian function below the surface mixed layer (*see* Fig. 2a). The three Gaussian parameters ($P_{max}$, $z_m$, $\sigma$) can characterize the SCM phenomenon. Thus, $z_m$ is the depth of the maximum chlorophyll (the peak of the bell shape), and $\sigma$ is the standard deviation of Gaussian function, which controls the width of the SCML. The upper and lower boundary of SCML can be defined as $z_m$-$\sigma$ and $z_m$+$\sigma$, respectively, which are at the depths where there is the balance between phytoplankton growth and losses and thus reflect the physical-biological ecosystem dynamics associated with SCML (Beckmann and Hense, 2007; Gong et al., 2015). To explore the SCML in stratified waters, we assume the surface mixed layer is shallower than the upper boundary of the SCML, i.e., $z_s$<$z_m$-$\sigma$. Examples of the piecewise function that reasonably well fitted to vertical chlorophyll profiles in the northern South China Sea (SCS) can be found in Gong et al. (2014).

The piecewise function approximation (Eq. 6) was evaluated and justified through numerical simulation of the nutrient-phytoplankton system (Eqs. 1-2), which is solved with a semi-implicit time stepping scheme. The vertical resolution is uniform (2 m), extending down to 200 m. We assumed a small uniformly distributed concentration of phytoplankton ($P(z,0)$=0.1 mg m$^{-3}$) and nitrogen ($N(z,0)$=0.1 mmol N m$^{-3}$) as the

initial conditions and run the model until in converge to a steady state (The modeling results show that the steady state has no relationship with the initial values of phytoplankton and nitrogen). We use the biologically reasonable parameter values given in Table 1 to represent the system at Station SEATS (South East Asia Time Series) in the northern SCS. Thus, the specific (calibrated) model solution is

considered as an example to obtain the analytic solutions of nitracline.

Fig. 3 shows the numerically simulated equilibrium distributions of nitrogen, light, and chlorophyll. In addition, the simulated vertical profile of chlorophyll is fitted well by the piecewise function of chlorophyll using the least square method (Fig. 3). Many numerical solutions of the nutrient-phytoplankton system have reproduced the vertical

chlorophyll profile with the SCML (Fennel and Boss, 2003; Huisman et al., 2006; Ryabov et al., 2010). Thus, analogous to the study by Klausmeier and Litchman (2001), we incorporate the piecewise function (Eq. 6) to the nutrient-phytoplankton system (Eqs. 1-2) at steady state to examine the roles of the SCML in reshaping the nitracline. We note that the useful delta function approximation in Klausmeier and

Litchman (2001) was verified by both simulation and rigorous mathematics (Du and Hsu, 2008a, b). As presented above, the assumption of the piecewise function approximation is physically practical.

*2.2 Definition of the nitracline*

The vertical distributions of nitrate often exhibit a strong gradient in depth (the

nitracline), but the feature of nitracline (depth, steepness) is variable in euphotic zones due to the combined effect of physical and biological processes.

Many studies define the nitracline depth as the location where the maximum vertical gradient in nitrate concentrations occurs (Eppley et al., 1979; Bahamón et al., 2003; Wong et al., 2007; Beckmann and Hense, 2007; Martin et al., 2010). To

measure the defined depth, a high vertical resolution of nitrate concentrations is needed and this is a big technical challenge existing for a long time. Thus, some definitions were also proposed to make the depth measurable. For example, one

definition is the depth where the nitrate concentration reaches a prescribed concentration, e.g., 0.05, 0.1, 1.0, or 12 mmol N m$^{-3}$ (Cullen and Eppley, 1981; Koeve et al., 1993; Martin and Pondaven, 2003). Some studies choose it to be the first depth where nitrogen is detectable (e.g., 0.05 or 0.1 mmol N m$^{-3}$) (Cermeno et al., 2008; Hickman et al., 2012) or where the nitrogen concentration exceeds the mixed layer value by a prescribed concentration difference (e.g., 0.05 mmol N m$^{-3}$) (Laanemets et al., 2004). Significant differences exist between these defined depths, i.e., the depth of maximal nitrate gradient was found to be deeper by 10 m from the first depth where nitrate can be detected (Eppley et al., 1978), while the nitrate gradient at the first detectable depth of nitrate is nearly zero (Cermeno et al., 2008).

With the development of nearly continuous nitrate profile measurement using the In Situ Ultraviolet Spectrophotometer (ISUS) optical nitrate sensor (Johnson and Coletti, 2002; Johnson et al., 2010), the detection of the maximum nitrate gradient could be more accurate than before. In this study, we adopt the location of the maximum nitrate gradient ($\max(dN/dz)$) in the euphotic zone as the nitracline depth ($z_n$), which can be expressed by $\frac{d^2N}{dz^2}\big|_{z_n} = 0$ and $\frac{d^3N}{dz^3}\big|_{z_n} < 0$.

Below the surface mixed layer, the steady-state version of Eq. (2) reduces to $\gamma\mu_m \min(f(I), g(N))P - \gamma\alpha\varepsilon P = K_{v2}\frac{d^2N}{dz^2}$. Thus, according to our model approach (Eq. 2) the nitracline depth where $\frac{d^2N}{dz^2} = 0$ represents a balance between the nutrient uptake and the recycling of phytoplankton loss, i.e., $\mu_m\min(f(I), g(N))=\alpha\varepsilon$.

In this study the nitracline steepness is defined as the nitrate gradient at the nitracline depth ($\frac{dN}{dz}\big|_{z_n}$) (Laanemets et al., 2004; Aksnes et al., 2007).

2.3 Data sources

The nitrate profiles were obtained from the ISUS measurement at the SEATS station during the CHOICE-C 2012 summer cruise. Nine casts were conducted during

Aug. 6-7, 2012. The raw ISUS nitrate data, which employed temperature-compensation, were first calibrated by the AutoAnalyzer 3 (AA3), and then smoothed to remove noise. The sampling frequency was set at 5 Hz and the raw data were thus smoothed with a 25-point moving average in the surface mixed layer, a 5-point moving average in the SCML, and a 15-point moving average below the SCML. The data were then interpolated by a cubic spline function. The corresponding temperature was obtained from Conductivity-Temperature-Depth (CTD) measurements. Overall, nine sets of profiles are available to examine our analytical solutions.

## 3 Results

### 3.1 Relations between nitracline and SCML

### 3.1.1 Nitracline depth and SCML

At steady state, multiplying Eq. (1) by $\gamma$ then adding Eqs. (1) and (2) leads to:

$$(\alpha-1)\varepsilon P - w\frac{dP}{dz} + \frac{d}{dz}\left(K_v(z)\frac{dP}{dz}\right) + \frac{1}{\gamma}\frac{d}{dz}\left(K_v(z)\frac{dN}{dz}\right) + \frac{N_{in}(z)}{\gamma} = 0 \qquad (7)$$

By substituting the expression of eddy diffusivity (Eq. 4) and the fitted, depth dependent function of chlorophyll ($P(z)$, Eq. 6) into Eq. (7), we obtain the diffusive term of nitrate below the surface mixed layer, that is,

$$K_{v2}\frac{d^2N}{dz^2} = \left[-\frac{K_{v2}}{\sigma^2}\left(\frac{z-z_m}{\sigma}\right)^2 - \frac{w}{\sigma}\left(\frac{z-z_m}{\sigma}\right) + \frac{K_{v2}}{\sigma^2} + (1-\alpha)\varepsilon\right]\gamma P \qquad z_s < z < z_b \qquad (8)$$

Letting $d^2N/dz^2 = 0$ in Eq. (8), for $P>0$ one gets

$$-\frac{K_{v2}}{\sigma^2}\left(\frac{z-z_m}{\sigma}\right)^2 - \frac{w}{\sigma}\left(\frac{z-z_m}{\sigma}\right) + \frac{K_{v2}}{\sigma^2} + (1-\alpha)\varepsilon = 0 \qquad z_s < z < z_b \qquad (9)$$

Solving this quadratic equation of depth $z$, we obtain the depths $z_{n1}$ and $z_{n2}$,

$$
\begin{cases}
z_{n1} = z_m - \dfrac{w\sigma^2}{2K_{v2}} - \sqrt{\dfrac{w^2\sigma^4}{4K_{v2}^2} + \dfrac{(1-\alpha)\varepsilon\sigma^4}{K_{v2}} + \sigma^2} \\[4mm]
z_{n2} = z_m - \dfrac{w\sigma^2}{2K_{v2}} + \sqrt{\dfrac{w^2\sigma^4}{4K_{v2}^2} + \dfrac{(1-\alpha)\varepsilon\sigma^4}{K_{v2}} + \sigma^2}
\end{cases}
\tag{10}
$$

Taking the derivative of Eq. (8) with respect to depth $z$, we get $K_{v2}\dfrac{d^3N}{dz^3} = \left[ -\dfrac{2K_{v2}}{\sigma^4}(z-z_m) - \dfrac{w}{\sigma^2} \right]\gamma P - K_{v2}\dfrac{d^2N}{dz^2}\dfrac{z-z_m}{\sigma^2}$. Obviously, at depth $z_{n1}$, $d^3N/dz^3 > 0$, and at depth $z_{n2}$, $d^3N/dz^3 < 0$. That is, $z_{n2}$ is the location of maximum nitrate gradients. We obtain that the nitracline depth refers to the depth $z_{n2}$, i.e.,

$$
z_n = z_{n2} = z_m - \frac{w\sigma^2}{2K_{v2}} + \sqrt{\frac{w^2\sigma^4}{4K_{v2}^2} + \frac{(1-\alpha)\varepsilon\sigma^4}{K_{v2}} + \sigma^2}
\tag{11}
$$

Particularly, Eq. (9) becomes a linear function of depth $z$ when the second order item coefficient ($K_{v2}/\sigma^4$) is zero, thus has only one solution. In fact, in typical stratified waters the diffusivity below the surface mixed layer ($K_{v2}$) is $1\text{-}9*10^{-5}$ $m^2\,s^{-1}$, and the thickness of SCML ($2\sigma$) is from several meters to tens of meters (Cullen, 2015), thus, $K_{v2}/\sigma^4$ (values from $8.64*10^{-9}$ to $7.78$ $m^{-2}\,s^{-1}$) can be neglected in some cases. When

$K_{v2}/\sigma^4 \to 0$, for non-zero sinking velocity we get one solution from Eq. (9),

$$
z_n = z_m + \frac{(1-\alpha)\varepsilon\sigma^2}{w} \qquad w \neq 0
\tag{12}
$$

Both Eqs. (11) and (12) show that the nitracline depth is located below the SCML depth, i.e., $z_n > z_m$ (Fig. 1). A numerical study in weak vertical mixing environments showed a similar result (Beckmann and Hense, 2007). Note that the SCML represents

the SBML in our model. In some oligotrophic oceans, the SCML will be deeper than the SBML due to the effect of photoacclimation on the vertical distribution of chlorophyll (Fennel and Boss, 2003). For example, Li et al. (2015) showed that the modeled maximum nitrate gradient (nitracline) occurred below the depth of SCML in the northern SCS, and then we can deduce that the nitracline depth is also deeper than

the depth of SBML. In the Mediterranean Sea, Bahamón et al. (2003) found that the

nitracline occurred below the depth of SCML at 88% of the stations (50 out of 57 stations). As well known, the SCML in the Mediterranean is often due to photoacclimation. It is not surprising that the other 12% of the stations (7 out of 57) in the Mediterranean Sea are against $z_n > z_m$ because photoacclimation leads to a much deeper SCML than the SBML. Thus, we conclude that the nitracline depth is located below the SBML depth.

*3.2.2 Nitracline steepness and SCML*

To illustrate the relationship between the nitracline steepness and the SCML, by integrating Eq. (8) from depth $z_n$ to $z_b$ and using the assumption for phytoplankton at the bottom boundary, i.e., $P \rightarrow 0$ for $z \rightarrow z_b$ (Eq. 5), we get

$$\frac{dN}{dz}\bigg|_{z_n} = \frac{dN}{dz}\bigg|_{z_b} + \left(\frac{z_n - z_m}{\sigma^2} + \frac{w}{K_{v2}}\right)\gamma P\big|_{z_n} - \frac{(1-\alpha)\varepsilon\gamma}{K_{v2}}\int_{z_n}^{z_b} Pdz \qquad (13)$$

This equality indicates that the nitracline gets steeper as the distance between the depths of nitracline layer and SCML is increased.

Incorporating Eq. (11) into Eq. (13) leads to

$$\frac{dN}{dz}\bigg|_{z_n} = \frac{dN}{dz}\bigg|_{z_b} + \left(\sqrt{\frac{w^2}{4K_{v2}^2} + \frac{(1-\alpha)\varepsilon}{K_{v2}} + \frac{1}{\sigma^2}} + \frac{w}{2K_{v2}}\right)\gamma P\big|_{z_n} - \frac{(1-\alpha)\varepsilon\gamma}{K_{v2}}\int_{z_n}^{z_b} Pdz \quad (14)$$

Equation (14) indicates that the nitracline steepness is negatively related to the thickness of SCML.

*3.2 Analytical solutions for nitracline features*

*3.2.1 Depth of the nitracline*

By substituting the general Gaussian function for chlorophyll below the surface mixed layer (Eq. 6) into Eq. (1), we obtain the steady-state net growth rate of phytoplankton below the surface mixed layer:

$$\mu_m \min\left(f(I), g(N)\right) - \varepsilon = -K_{v2}\left(z - \left(z_m - \frac{w\sigma^2}{2K_{v2}}\right)\right)^2 \bigg/ \sigma^4 + w^2\big/4K_{v2} + K_{v2}\big/\sigma^2 \quad (15)$$

Let $z_0 = z_m - w\sigma^2/(2K_{v2})$ in the first term of the right-hand of Eq. (15). From the result given by Gong et al. (2015), we know that $z_0$ is the location of the maximum growth rate of phytoplankton (hereafter named as the depth of optimal growth), where the transition from nutrients limitation to light limitation occurs (i.e., $f(I)=g(N)$ at depth $z_0$).

Clearly, $z_n>z_0$ (Eq. 11). Hence, the growth of phytoplankton at the nitracline depth $z_n$ is limited by light, i.e., $\mu_m \min(f(I), g(N))\big|_{z_n} = \mu_m f(I)\big|_{z_n}$. In other words, the growth rate of phytoplankton at the nitracline depth is a function of the light level at the nitracline depth, $I(z_n)$. Thus, from Eqs. (11) and (15), we obtain the growth rate of phytoplankton at the nitracline depth, that is,

$$\mu_m f(I)\big|_{z_n} = \alpha\varepsilon \tag{16}$$

Note that the derivation of Eq. (16) only works when light and nutrient limitation (Blackman's law of limiting factors) is applied. Substituting the Michaelis-Menten form for $f(I)$ into Eq. (16), we have

$$I(z_n) = \frac{K_I}{\mu_m/\alpha\varepsilon - 1} \tag{17}$$

Rearranging Eq. (17), we find $\mu_m - \alpha\varepsilon = \alpha\varepsilon K_I/I(z_n)$. This equality indicates that the maximum rate of NPP, $(\mu_m-\alpha\varepsilon)P$, is inversely proportional to the light level at the nitracline depth, $I(z_n)$. Lande et al. (1989) found that higher maximum rates of population growth corresponded to shallower nitracline depths in the central North Atlantic.

Furthermore, insertion of Eq. (3) into Eq. (17) yields another expression of the nitracline depth:

$$z_n = \frac{1}{K_w} \ln \frac{I_0 (\mu_m/\alpha\varepsilon - 1)}{K_I} - \frac{K_c}{K_w} \int_0^{z_n} P dz \tag{18}$$

Note that Eq. (18) is obtained on the premise that the nitracline depth exists. This

equality shows that the nitracline depth is inversely proportional to the light attenuation coefficient of water ($K_w$), and it deepens logarithmically with increasing surface light intensity ($I_0$). It is noted that the nitracline depth has a negative relation with the self-shading of phytoplankton ($K_c \int_0^{z_n} P dz$).

Importantly, Eq. (18) predicts that the nitracline depth has no relation with subsurface diffusivity. Aksnes et al. (2007) also proposed a similar result that a shoaling nitracline per se cannot be taken as an unequivocal sign of increased upwelling, as well as eddy diffusion. However, this does not mean that fluid dynamics are unimportant in shaping vertical distribution of nitrate.

Equation (18) also indicates that both a higher recycling efficiency ($\alpha$) and a larger loss rate ($\varepsilon$) lead to a shallower nitracline, while the enhanced maximum growth rate of phytoplankton ($\mu_m$) moves the nitracline depth downward. Modeling results showed that the nitracline was shoaled by 24% (from 84 m upwards to 64 m) when both the recycling efficiency ($\alpha$, from 0.6 to 0.8) and the loss rate ($\varepsilon$, from 0.3 d$^{-1}$ to 0.4 d$^{-1}$) were increased by 33%. Accordingly, the predicted nitracline depth from Eq. (18) varied from 86 m to 71 m. Increasing $\mu_m$ by 33% (from 0.9 d$^{-1}$ to 1.2 d$^{-1}$) makes the simulated nitracline deepening slightly (from 84 m to 88 m), leading to the predicted nitracline depth changing from 86 m to 92 m. The experiments with varying parameter values indicate that the updated $z_n$ (based on the model runs) matches well the predicted $z_n$ of Eq. (18).

*3.2.2 Steepness of the nitracline*

In steady state, integrating Eq. (2) from the nitracline depth $z_n$ to the bottom boundary $z_b$, and considering the light limitation of phytoplankton growth below depth $z_n$ (Eq. 15), we obtain the nitrate gradient below the surface mixed layer,

$$\frac{dN}{dz}\Big|_{z_n} = \frac{dN}{dz}\Big|_{z_b} + \frac{1}{K_{v2}} \int_{z_n}^{z_b} \left(\alpha\varepsilon - \mu_m f(I)\right) \gamma P dz \tag{19}$$

This equality shows that the nitracline steepness enhances with increasing nitrate

gradient at the bottom boundary ($\frac{dN}{dz}\big|_{z_b}$) which depends on the intensity of nitrate

intrusion from below. The vertical diffusion negatively influences the nitracline

steepness. The modeled time-series distributions of nitrate gradients and diffusive

nitrate fluxes in the northern SCS and the upstream Kuroshio Current showed similar

results (Li et al., 2015). Beckmann and Hense (2007) conducted sensitivity analysis of

both vertical diffusivity and nutrient concentration at the bottom boundary to examine

the vertical phytoplankton and nutrient profiles in oligotrophic oceans and their

numerical results support the relations presented in Eq. (19).

*3.3 Analytical solutions for SCM characteristics*

Similar to methods used by Gong et al. (2015), the piecewise function for vertical

chlorophyll profile (Eq. 6) was incorporated into the nutrient-phytoplankton model

(Eqs. 1-2) at steady state to derive the three SCM characteristics (SCML thickness, its

depth and intensity).

For $z = z_m$ and $z = z_m + \sigma$, the net growth rate of phytoplankton (Eq. 15) can be

respectively expressed as follows:

$$\mu_m f(I)|_{z=z_m} - \varepsilon = K_{v2}/\sigma^2 \tag{20}$$

$$\mu_m f(I)|_{z=z_m+\sigma} - \varepsilon = -w/\sigma \tag{21}$$

By substituting the growth limitation function for light to Eq. (20) or Eq. (21), we

obtain the expression of parameter $z_m$, i.e.,

$$z_m = \frac{1}{K_w} \ln\left[\left(\frac{\mu_m}{\varepsilon + K_{v2}/\sigma^2} - 1\right)\frac{I_0}{K_I}\right] - \frac{K_c}{K_w}\int_0^{z_m} P dz \tag{22}$$

or

$$z_m = \frac{1}{K_w} \ln\left[\left(\frac{\mu_m}{\varepsilon - w/\sigma} - 1\right)\frac{I_0}{K_I}\right] - \frac{K_c}{K_w}\int_0^{z_m+\sigma} P dz - \sigma \tag{23}$$

Subtracting Eqs. (22) and (23), and rearranging, we obtain the expression of parameter σ:

$$\left(\frac{\mu_m}{\mu_m - \varepsilon + w/\sigma} - 1\right) \exp^{K_w \sigma + K_c \int_{z_m}^{z_m+\sigma} P dz} = \frac{\mu_m}{\mu_m - \varepsilon - K_{v2}/\sigma^2} - 1 \tag{24}$$

Neglecting terms including self-shading of phytoplankton ($K_c$) in Eqs. (22-24), both the analytical solutions of the depth and thickness of SCML are the same as the results presented in Gong et al. (2015). The self-shading effect of phytoplankton plays an important role in vertical pattern of chlorophyll (Shigesada and Okubo, 1981; Klausmeier and Litchman, 2001; Beckmann and Hense, 2007). In line with common sense, our analytic results indicate that a higher self-shading of phytoplankton negatively influences the depth and thickness of the SCML (Eq. 22 and Eq. 24), having similarly effect as an increasing light attenuation coefficient of water, $K_w$.

The expression of the SCML intensity is different from the results presented in Gong et al. (2015). Integrating Eq. (7) from the surface of water to the bottom of surface mixed layer ($z_s$), and from the bottom of surface mixed layer to the base of our model domain ($z_b$) respectively, gives:

$$(1-\alpha)\varepsilon\gamma P_0 z_s = K_{v2} \frac{dN}{dz}\Big|_{z_s+0} + N_{inML} \tag{25}$$

$$(1-\alpha)\varepsilon\gamma\lambda h = K_{v2} \frac{dN}{dz}\Big|_{z_b} - K_{v2} \frac{dN}{dz}\Big|_{z_s+0} \tag{26}$$

Parameter $\lambda h$ (mg/m$^2$) is assumed as the integrated chlorophyll concentration below the surface mixed layer. Based on the properties of Gaussian function, $\lambda$ can be expressed as $\lambda = \Phi\left(\frac{z_b - z_m}{\sigma}\right) - \Phi\left(\frac{z_s - z_m}{\sigma}\right)$, where $\Phi\left(\frac{z_b - z_m}{\sigma}\right)$ and $\Phi\left(\frac{z_s - z_m}{\sigma}\right)$ can be obtained from the standard normal table. According to the property of Gaussian function, we have $0.68 < \lambda < 1.00$, $z_s < z < z_b$, under the assumption of $z_s < z_m - \sigma$.

Adding Eqs. (25) and (26) yields:

$$(1-\alpha)\varepsilon\gamma(\lambda h + P_0 z_s) = K_{v2}\frac{dN}{dz}\Big|_{z_b} + N_{inML} \qquad (27)$$

Nitrogen input to the surface mixed layer ($N_{inML}$) causes an increase of surface chlorophyll concentration (Eq. 25). Hence, the total chlorophyll in stratified water columns ($\lambda h + P_0 z_s$) increases with increasing $N_{inML}$ (Eq. 27), which has also been predicted by the numerical study of Mellard et al. (2011) and supported by the experimental test of Mellard et al. (2012).

Because recycling processes are assumed to not immediately convert dead phytoplankton back into dissolved nutrients below the surface mixed layer, i.e., $\alpha \neq 1$, the total chlorophyll concentration below the surface mixed layer and the intensity of SCML can be respectively expressed as:

$$\lambda h = \frac{K_{v2}}{(1-\alpha)\varepsilon\gamma}\frac{dN}{dz}\Big|_{z_b} + \frac{N_{inML}-(1-\alpha)\varepsilon P_0 z_s}{(1-\alpha)\varepsilon\gamma} \qquad (28)$$

$$P_{\max} = \frac{1}{\lambda\sqrt{2\pi}\sigma}\left(\frac{K_{v2}}{(1-\alpha)\varepsilon\gamma}\frac{dN}{dz}\Big|_{z_b} + \frac{N_{inML}-(1-\alpha)\varepsilon P_0 z_s}{(1-\alpha)\varepsilon\gamma}\right) \qquad (29)$$

The integrated chlorophyll concentration below the surface mixed layer ($\lambda h$) and the intensity of SCML ($P_{\max}$) are influenced by $N_{inML}$ positively and by $P_0$ negatively (Eqs. 28-29). That is to say, the influence of nitrate input to the surface mixed layer on the SCML intensity (also on the integrated chlorophyll concentration below the surface mixed layer) is non-linear. Hence, their changes ($\lambda h$ and $P_{\max}$) with varying $N_{inML}$ cannot be obtained from the steady-state solutions straight forwardly, depending on the specific parameter combinations in the model. For example, $\lambda h$ and $P_{\max}$ decrease when increasing nutrient enrichment directly to the surface mixed layer in the ecosystem given by Mellard et al. (2011), while they are nearly unchanged in oligotrophic oceans (Varela et al., 1994).

Our results (Eqs. 28-29) also show that enhanced subsurface diffusivity ($K_{v2}$) increases the integrated chlorophyll concentration and the intensity of the SCML ($\lambda h$ and $P_{\max}$), as a result of a higher nitrate flux ($K_{v2}n$). Physical upward transport of

nitrate across the bottom of nitracline is indeed the main nitrogen source for NPP in the euphotic zone (Ward et al., 1989).

## 4 Discussion

### 4.1 Light effects on nitracline with SCML

We now examine how the steady state nitracline in relate to SCML depends on light availability, especially light level at the nitracline depth.

Substituting Eq. (17) to Eq. (28) and rearranging, we have

$$\frac{K_{v2}}{\lambda h + P_0 z_s}\frac{dN}{dz}\Big|_{z_b} + \frac{N_{inML}}{\lambda h + P_0 z_s} = \varepsilon\gamma - \frac{\mu_m\gamma}{K_I/I(z_n)+1} \tag{30}$$

This equality indicates that the light level at the nitracline depth, $I(z_n)$, is positively related to the integrated chlorophyll concentration in the whole water column, $\lambda h + P_0 z_s$. we can derive from Eq. (3) that the nitracline depth ($z_n$) is inversely related to integrated chlorophyll. This inverse relationship has been observed in many regions. In southern California coastal waters, the phytoplankton standing stock and its primary production rate were positively related to the reciprocal nitracline depth (Eppley et al., 1978; Eppley et al., 1979). Bahamón et al. (2003) found that larger depth-integrated chlorophyll with an average deeper SCML and nitracline (~129m, ~136m, respectively) occurred in the Western Sargasso, Central Sargasso and Eastern Atlantic, compared with that in the Canary Current zone.

The nitracline depth deepens with increasing surface light intensity but with decreasing light attenuation coefficients ($K_w$ and $K_c$). These results were consistent with observations, e.g., Letelier et al. (2004) found the depth of the nitracline to coincide with an isolume, a depth of constant light level in the North Pacific Subtropical Gyre.

The predicted effect of surface light intensity and light attenuation coefficient on the nitracline depth (Eq. 18) implies that the nitracline depth in stratified waters may have seasonal variations. In the North Pacific Subtropical Gyre, Letelier et al. (2004)

found that the nitracline depth differences between winter and summer disappeared when nitrate concentrations were plotted against light level in the water column. Aksnes et al. (2007) found that the seasonal pattern of nitracline depth was governed by seasonality in light attenuation coefficient, rather than in surface light intensity. Particularly, the inverse proportional relationship between the nitracline depth and light attenuation coefficient (Eq. 18) has also been derived from a steady-state model by Aksnes et al. (2007), which is consistent with observations in the coastal upwelling region off Southern California (Aksnes et al., 2007). Tiera et al. (2005) found a significant positive correlation between the nitracline depth and the depth of 1% surface light intensity (the proportion of reciprocal light attenuation coefficient) in the Eastern North Atlantic Subtropical Gyre. Bahamón et al. (2003) showed that the nitracline depth remained relatively constant around 1% surface light intensity depth in Western Sargasso.

The nitracline steepens with a higher light attenuation coefficient ($K_w$ and $K_c$) due to $K_w$ and $K_c$ negatively influencing SCML thickness (Eqs. 14 and 25). Numerical modeling showed that a higher $K_w$ leads to a thinner SCML and thus a steeper nitracline layer (Beckmann and Hense, 2007). Aksnes et al. (2007) also found that the fluctuations in the nitracline steepness were positively correlated with the fluctuations in reciprocal Secchi depth (i.e., light attenuation coefficient) in the upwelling area off the coast of the Southern California. We further point out that the nitracline steepness almost stays constant when changing surface light intensity ($I_0$), because surface light intensity has no relation to the SCML thickness (Eq. 25). The sensitivity analysis of a one-dimensional (vertical) model showed that the vertical nutrient profiles were almost paralleling with each other when increasing surface light intensity (Beckmann and Hense, 2007).

The inverse effects of light attenuation coefficient on the nitracline steepness and its depth imply that the nitracline becomes steeper as the nitracline shoals. Aksnes et al. (2007) found this consistent pattern in the upwelling area off the coast of the Southern California.

*4.2 In presence of surface nutrient input*

Current evidences and modeling analyses suggest that climate warming will increase ocean stratification, and hence reduce nutrient exchange between the ocean interior and the upper mixed layer (Cermeno et al., 2008; Chavez et al., 2011). Therefore, nutrient input directly to the euphotic layer due to atmospheric deposition may become a relatively more important nutrient supply mechanism to the euphotic

layer (Mackey et al., 2010; Okin et al., 2011; Mellard et al., 2011). However, few model studies (e.g., Mellard et al. 2011) have explored the influences of external surface nutrient supply on vertical phytoplankton distribution.

Observations show that an inter-zone exists between the transition of the surface mixed layer and the deep layer, where the nutrient gradient equals nearly zero

$\left. \dfrac{dN}{dz} \right|_{z_s+0} = 0$ (Fig. 3), leading to the solution in Eq. (26). It follows that the total chlorophyll in the surface mixed layer depends on the surface nutrient supply ($N_{inML}$) (Eq. 26). In this case, if $N_{inML}$ is negligible, Eq. (26) degenerates to

$$(1-\alpha)\varepsilon\gamma P_0 z_s = 0 \tag{31}$$

In this case, the dead phytoplankton in surface mixed layer must be fully recycled,

i.e., $\alpha=1$, in order to sustain the positive chlorophyll concentration ($P_0>0$). In other words, if the dead phytoplankton cannot be fully recycled in the surface mixed layer, external nutrient supply to the layer is needed to fuel the growth of phytoplankton. Thus, the term, external nutrient supply to the surface mixed layer, should be included in the system equations at steady state to make a positive surface chlorophyll

concentration. Numerical results by Mallard et al. (2011) also showed that phytoplankton populations can grow in the mixed layer and in the deep layer together, when there is nutrient input directly to the mixed layer. However, a surface nutrient source is not a necessary term for a model approach where dissolved organic matter and detritus are explicitly resolved (Beckmann and Hense, 2007).

Accordingly, we treat the vertical phytoplankton distribution as a piecewise

function, comprised by a linear function in the surface mixed layer and a Gaussian function below, which is more realistic than the general Gaussian function. The assumption of the piecewise function for phytoplankton is also consistent with the assumption of piecewise vertical diffusivity. For simplicity, we assume that the transition layer between the surface mixed layer and the deep one is infinitely thin, and the chlorophyll is continuous within the transition layer. By assuming the SCML depth is significantly deeper than the base of the surface mixed layer, we obtain the steady state solutions for the SCML depth and thickness, similar to the solutions using the general Gaussian function. However, the intensity of the SCML is affected by surface nutrient supply with an associated positive increase in phytoplankton concentration.

*4.3 SCML trapping Nutrient*

Observations and numerical simulations showed that SCML played a role as a nutrient trap in some regions, restricting the diffusive flux of nitrates to the surface mixed layer (Anderson, 1969; Klausmeier and Litchman, 2001; Navarro and Ruiz, 2013).

From Eq. (10), we know $z_s < z_{n1} < z_0 - \sigma < z_m - \sigma$ (Fig. 1). That is, the SCML occurred below depth $z_{n1}$. For $z_m < z_n$ (Eq. 10), we know that the upward diffusive nitrate is sufficient to sustain the fast growth of phytoplankton in the lower part of the SCML ($z_m < z < z_m + \sigma$). To explore the SCML restricting nitrates into the surface mixed layer, next, we examine if the nitrate concentration at depth $z_{n1}$ above the upper boundary of the SCML ($z_{n1} < z_m - \sigma$) is depleted.

According to the definition of the depth $z_0$ (where $f(I) = g(N)$ holds) and $z_0 > z_{n1}$ (Eq. 10, Fig. 1), we know that the growth of phytoplankton at depth $z_{n1}$ is nitrate-limited, i.e., $\mu_m \min\left(f(I), g(N)\right)\big|_{z_{n1}} = \mu_m g(N)\big|_{z_{n1}}$. From Eq. (14), we get that at depth $z_{n1}$, the growth rate of phytoplankton equals the recycling efficiency of dead phytoplankton, i.e., $\mu_m g(N)\big|_{z_{n1}} = \alpha\varepsilon$. Inserting the Michaelis-Menten form for $g(N)$

into this equality yields: $N(z_{n1}) = K_N/(\mu_m/\alpha\varepsilon - 1)$. Phytoplankton maximum growth rates ($\mu_m$) of 0.2 to 1 per day are typical in optical environmental conditions (Banse, 1982; Timmermans et al., 2005). We choose 0.5 per day to illustrate the result. Loss rate ($\varepsilon$), although not well documented, is often quoted as 10% of the growth rate (Parsons et al., 1984). A reasonable choice for the remineralization efficiency seems to be $\alpha$=0.5 (Huisman et al., 2006). The typical value of half-saturation constants for nitrate ($K_N$) is between 0.1 and 0.7 mmol N m$^{-3}$ in oceans (Eppley et al., 1969). We adopt 0.4 mmol N m$^{-3}$. Thus, we obtain that the nitrate concentration at depth $z_{n1}$, $N(z_{n1})$, is equal to 0.03 mmol N m$^{-3}$, a value lower than the detection limit, indicating the depletion of nitrate above depth $z_{n1}$.

Because the SCML acts as a nutrient barrier, it is easy to understand that the rate of NPP in the SCML ($(\mu_m \min(f(I), g(N)) - \alpha\varepsilon)P$, $z_m - \sigma < z < z_m + \sigma$) is positively related to upward nitrate flux that is trapped. This condition can simply be derived by integrating Eq. (2) vertically at steady state, i.e., $\int_{z_m-\sigma}^{z_m+\sigma} (\mu_m \min(f(I), g(N)) - \alpha\varepsilon)\gamma P dz = K_{v2} \, dN/dz \big|_{z_m-\sigma}^{z_m+\sigma}$. This result suggests that, the production within the SCML is fuelled mainly, by nitrate and is thus NPP. Because at the nitracline depth the gross growth rate $\mu_m\min(f(I), g(N))$ equals the recycling of dead phytoplankton $\alpha\varepsilon$, for the constant $\alpha\varepsilon$ we assumed, within the nitracline layer ($z_{n1}<z_m$-$\sigma$ and $z_m<z_{n2}$) the nitrate uptake by phytoplankton has to be supplied by the vertical diffusion. Observations also showed that most of the primary production in SCML was supported by nitrate from vertical diffusion, with an average $f$-ratio (i.e., relative contribution of the nitrate uptake to the total nitrogen uptake) of 0.74±0.26 during early summer in Canadian Arctic waters (Martin et al., 2012).

### 4.4 Vertical profile of nitrate gradients

From the monotonicity of the quadratic function of depth $z$ in the left-hand of Eq. (9), we know that $d^2N/dz^2 < 0$ when $z_s<z<z_{n1}$ and $z_{n2}<z<z_b$, but $d^2N/dz^2 > 0$ when $z_{n1}<z<z_{n2}$. In other words, the vertical gradient of the nitrate concentration ($dN/dz$)

decreases with depth on the interval ($z_s$, $z_{n1}$), while increases on the interval ($z_{n1}$, $z_{n2}$),

and then decreases on the interval ($z_{n2}$, $z_b$). If we consider the distribution of vertical

nitrate gradients as continuous across the base of the surface mixed layer, then we get

$dN/dz<0$ for $z_s<z<z_{n1}$ under the assumptions of the uniform nitrate distribution within

the surface mixed layer (i.e., $dN/dz=0$, $0<z<z_s$). The schematic of vertical profiles of

nitrate gradients and chlorophyll concentrations in stratified waters is shown in Fig. 1.

The negative gradient of nitrate below the surface mixed layer ($dN/dz<0$ for

$z_s<z<z_{n1}$) indicates that the nitrate concentration decreases with depth on the interval

($z_s$, $z_{n1}$). This decreasing nitrate feature has rarely been observed by traditional

measurements probably due to the technique-limited low resolution. Some float data

showed this feature in vertical nitrate profiles, for example, Sakamoto et al. (2009)

found it at depths below the base of surface mixed layer (~45-50 m) by the ISUS

temperature-compensated data at an eastern Pacific oligotrophic station. Our in situ

time series measurements using the ISUS at SEATS station also showed this

decreasing feature at depths ~25-30 m (Fig. 4). We note that this decreasing nitrate

feature will disappear in our derivation if the subsurface vertical diffusion is too weak

(Eq. 12) or the surface mixed layer is deeper than depth $z_{n1}$. Simulating results showed

that the negative gradient of nitrate becomes smaller with increasing the sinking

velocity (w) and the recycling efficiency ($\alpha$). This implies that the negative gradient is

likely the result of the consumption of nitrate by phytoplankton exceeding the supply

of nitrate.

*4.5 Limitation and application*

The model in this study integrates a number of physical, chemical, and biological

processes that act together to determine the vertical distribution of phytoplankton and

nitrate, under the assumption that the system is strictly vertical and in steady state. A

few processes such as oxygen status, photoacclimation, luxury uptake of nutrients,

phytoplankton motility, concentration-dependent-herbivory, and depth-dependent

herbivory are not included, although they can affect the vertical distribution of

phytoplankton and nitrate. Detritus, dissolved organic matter, and zooplankton are not

included explicitly, and all loss processes, except sinking, are set to be linearly proportional to phytoplankton. The sinking velocity of phytoplankton is assumed independent of density gradients. Further, the vertical transport of nutrients is only by eddy diffusion in our model; in reality, nutrients can be supplied by many processes (turbulence, internal waves, storms, slant-wise and vertical convection), especially by upwelling (Katsumi and Hitomi, 2003; Aksnes et al., 2007).

In this study, the sinking velocity of phytoplankton is set independent of nitrate concentration. Vertically-varying sinking velocity have been observed as physiological response to variations in light or nutrient levels (Steele and Yentsch, 1960; Bienfang and Harrison, 1984; Richardson and Cullen, 1995). The sinking velocity reduces with decreased light level and with increased nutrient concentration, and the resulting divergence in sinking velocity can be large enough to affect the location of the phytoplankton particle maximum. However, numerical results given by Fennel and Boss (2003) showed analytically that the divergence of the sinking rate contributes to the location of the SBM layer in a significant way only when the divergence in sinking rates occurred above the compensation depth in stable, oligotrophic environments. They also derived that in stable, oligotrophic environments with a predominance of small cells, the biomass maximum is located at the depth where growth and losses are equal, leaving few influence by sinking divergence.

It is worth pointing out that, in extreme oligotrophic regions, the SCML is very deep and attributable mostly to photoacclimation of chlorophyll content rather than to a peak of biomass (Steele 1964; Fennel and Boss, 2003; Cullen, 2015). The process of photoacclimation is also important for the nutrient-phytoplankton system (with stratified conditions) we focused on. To explore the influence of photoacclimation on the nitracline, we parameterized Chl:C using the mathematical description by Cloern et al. (1995), i.e., $\text{Chl:C} = 0.003 + 0.0154 \exp(0.050T) \exp(-0.059I)\mu'$. That is, Chl:C is the ratio of Chl a to C in phytoplankton growth at steady state under defined temperature $T$ ($°C$), daily irradiance $I$ (mol quanta m$^{-2}$ d$^{-1}$), and at nutrient-limited

growth rate $\mu'$ ($\mu' = N/(K_N + N)$). The ratio Chl:C increases when temperature $T$ or

nitrogen concentration $N$ increases, while decreases with increasing daily irradiance $I$.

Let R=Chl:C, then the nitrogen content of phytoplankton can be written as

$\gamma = 1/(6.625 \ast 12 \ast R)$, corresponding to a C:N ratio of 6.625 and a carbon atomic mass of

12. From the expression of nitracline depth (Eq. 18), we know that the ratio Chl:C has

no influence on the nitracline depth. While the nitracline steepness increases with

increasing parameter $\gamma$ (Eq. 19). In other words, the nitracline gets steeper with a

lower ratio Chl:C. Note that a certainly more realistic model would be one with

equations that explicitly resolve variations of the Chl a-to-carbon and

nitrogen-to-carbon ratio of phytoplankton.

The piecewise equation (Eq.6) can be used to mimic a large variety of vertical

chlorophyll profiles from coastal, upwelling, open oceans and high latitude waters

(Fig. 2). For example, for $z_s$>0, when the depth of surface mixed layer equals or is

deeper than the depth of SCML, the vertical profiles like Fig. 2b and 2c are often

found in well-mixed waters (Uitz et al., 2006). For $z_s$=0, the vertical distribution of

chlorophyll concentration (Fig. 2d) can be expressed by a Gaussian function, which is

usually found in coastal upwelling waters (Xiu et al., 2008). Particularly, when $z_s$=0

and $z_m$=0, the surface bloom occurs (Fig. 2e). In general, the vertical profiles of

chlorophyll can be classified into two types, i.e., one peak distribution or uniform

distribution in large regions of lakes and oceans (Uitz et al., 2006; Lavigne et al.,

2015). Note that the skewed profiles of chlorophyll with a sharp SCM was not

considered in this study. The small-scale (1 m or less) vertical heterogeneity in

chlorophyll distribution has been shown to be common features in coastal waters

(Sullivan et al., 2010; Prairie et al., 2011; Durham and Stocker, 2012), named as thin

layer.

Choosing the values of model parameters represented the system in the northern

South China Sea (given in Table 1), we can retrieve the nitracline depth and steepness,

the optimal depth and the three SCM characteristics. To make calculation easy, we

neglect the term of self-shading by phytoplankton in the calculation, because a higher

self-shading parameter has the same effect as an increasing light attenuation coefficient by water. The calculated and observed values of these parameters are listed at Table 2. All these parameters calculated are in a reasonable range, although there are some discrepancies compared with observations. In fact, this is not surprising, considering that we assume a single phytoplankton group and neglect the microbial loop and the dynamics of the dissolved organic matter and detritus pools.

We stress that the analytical solutions of nitracline are valid only for estimates of $z_m$, $h$, and $\sigma$ that are consistent with the model's numerical steady-state solution. The numerical steady-state solution in turn depends on the combination of parameter values and on the forcing, boundary conditions. The approximations of $z_m$, $h$, and $\sigma$ are entirely conditioned by the modeling results and thus also depend on the combination of model parameter values. To combine the analytical steady state solutions with observed $z_m$, $h$, and $\sigma$ (as derived from vertical profiles of chlorophyll $a$ concentration) is only meaningful after model calibration (identifying a model solution that is in some agreement with the observed $z_m$, $h$, and $\sigma$).

## 5 Summary

We have presented a theoretical framework to investigate the interaction of phytoplankton and nutrient in stratified water column. A piecewise function for chlorophyll profiles comprising a linear function in the surface mixed layer and a Gaussian function below is assumed in the nutrient-phytoplankton model at steady state. A number of important findings are obtained under conditions of the model equations imposed.

In steady state, the nitracline is confined between two depths where the gross growth rate equals the recycling efficiency of dead phytoplankton, indicating that within the nitracline, nitrate consumption by phytoplankton has to be replenished by the upward flux of nitrate. This layer thereby is the major contributor to new primary production.

The nitracline depth locates below the SCML depth; both two depths deepen with

either increased surface light intensity or decreased light attenuation coefficient. The nitracline depth does not depend on the value of the subsurface diffusivity. The nitracline is steeper with a thinner SCML. The nitracline steepness is positively influenced by the light attenuation coefficient, yet, responds insignificantly to surface light intensity.

Our analytical solutions show that phytoplankton in the SCML acts as an efficient nutrient trap, filtering out the upward nitrate supply. The light level at the nitracline depth has a positive relation with the depth-integrated chlorophyll concentration in the whole water column and with the maximum rate of NPP, acting as the indicator of integrated NPP. The NPP is constructed from the model equations that rely in Blackman's law of limiting factor for the growth rate.

*Acknowledgements.* The authors thank State Key Laboratory of Marine Environmental Science, Xiamen University for providing ISUS nitrate and CTD data, especially acknowledge C. J. Du. We are very grateful to the Associate Editor (Jack Middelburg) and two reviewers (A. W. Omta and the other anonymous one) for their constructive and helpful suggestions. We also would like to thank Xiaohuan Liu, Yang Yu, and Xiaokun Ding for valuable advice and programming assistance. This work is funded in part by the National Key Basic Research Program of China 2014CB953700, and the National Nature Science Foundation of China (41406010, 41210008, and 91328202).

Figures

Figure 1

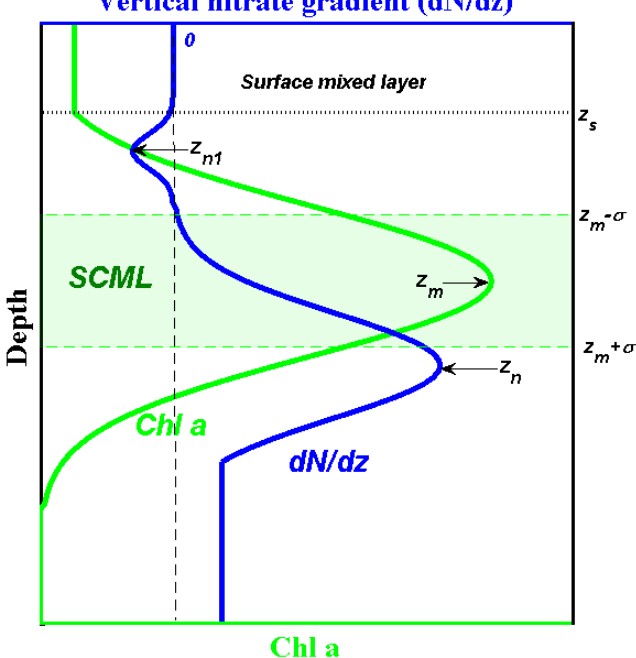

Fig. 1 Schematic picture of vertical profiles of nitrate gradient and chlorophyll (Chl a) in stratified water columns. (blue solid line is the vertical profile of nitrate gradient; green solid line is Chl a

concentration as a function of depth; red solid line represents the growth limitation by light, red dotted line by nitrate; horizontal green solid lines indicate the locations of the upper and lower SCML, $z_m$-σ, $z_m$+σ, respectively; horizontal black dotted line indicates the depth of the surface mixed layer, $z_s$; vertical dotted black line represents zero nitrate gradient; $z_{n1}$ and $z_n$ are the locations of extrame nitrate gradients, $z_n$ is the nitracline depth, and $z_m$ is the depth of maximum

chlorophyll concentration)

Figure 2

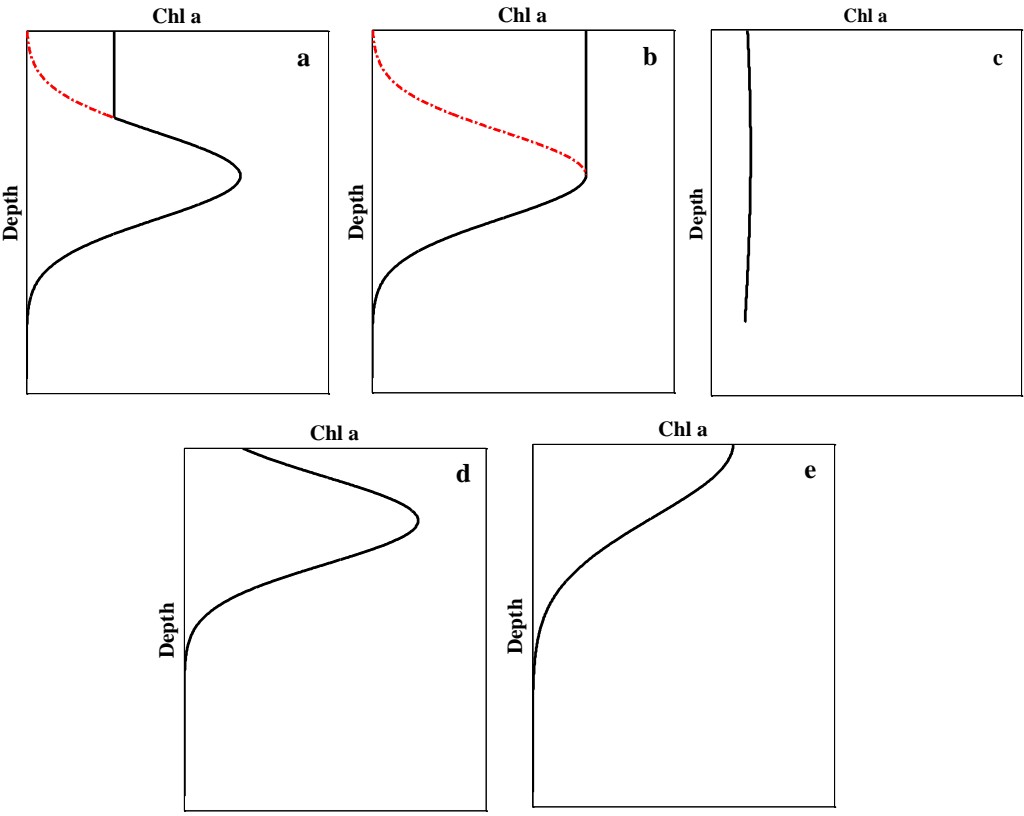

 Fig. 2 Examples of the vertical profiles of chlorophyll (black solid line). (red dotted lines represent the parts of Gaussian fitting curves, not the actual chlorophyll)

Figure 3

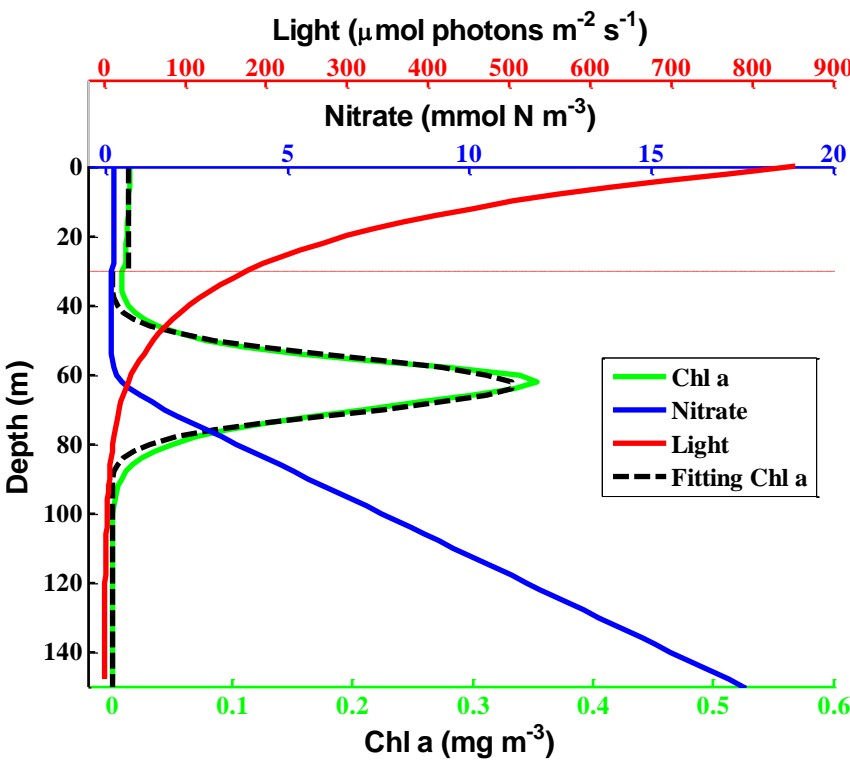

Fig. 3 Steady-state vertical distributions of chlorophyll, nitrate, and light determined by numerical solutions of Eqs. (1) and (2). Horizontal red dash-dotted line indicates the depth of the surface mixed layer. Black dash line represents the fitting curve of vertical chl a profile. The fitting

equation is $P = \begin{cases} 0.013 & 0 \le z \le 30 \\ 0.33\exp\left[-\dfrac{(z-63)^2}{2\times 9^2}\right] & 30 < z < 200 \end{cases}$.

Figure 4

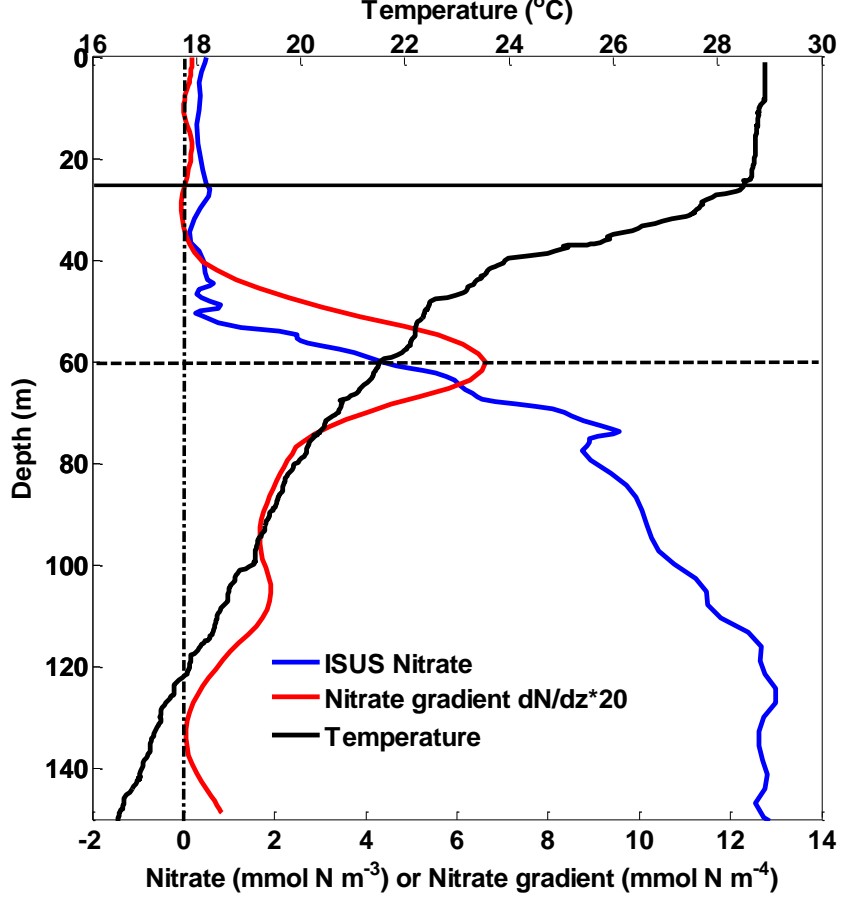

Fig. 4 Vertical nitrate gradient, ISUS nitrate and temperature at SEATS station (2012, cast 36) (horizontal line indicates the depth of the surface mixed layer, horizontal dotted line indicates the depth of nitracline, and the vertical dash-dotted line represents zero nitrate gradient).

Table 1 List of symbols and their values used in models at SEATS station in northern SCS

| Model parameters | Description (unit) | Values (range) |
|---|---|---|
| $I_0$ | Surface light intensity ($\mu mol\ photons\ m^{-2}\ s^{-1}$) | 900 (200-1700) [1, 2] |
| $K_w$ | Light attenuation coefficient of water ($m^{-1}$) | 0.052 [1, 3] |
| $K_c$ | Light attenuation coefficient of phytoplankton ($m^2\ (mmol\ N)^{-1}$) | 0.05 [1, 3] |
| $K_I$ | Half-saturation constant of light limited growth ($\mu mol\ photons\ m^{-2}\ s^{-1}$) | 40 [4] |
| $K_{v1}$ | Surface diffusivity (($\times 10^{-4})\ m^2\ s^{-1}$) | 2 [5] |
| $K_{v2}$ | Subsurface diffusivity (($\times 10^{-5})\ m^2\ s^{-1}$) | 5 [5] |
| $w$ | Sinking velocity of phytoplankton ($m\ d^{-1}$) | 1 [6] |
| $\varepsilon$ | Loss rate of phytoplankton ($d^{-1}$) | 0.3 [7] |
| $\alpha$ | Nutrient recycling coefficient (dimensionless) | 0.6 [7] |
| $K_N$ | Half-saturation constant of nutrient uptake ($mmol\ N\ m^{-3}$) | 0.4 [8] |
| $\mu_m$ | Maximum growth rate of phytoplankton ($d^{-1}$) | 0.9 [5, 7] |
| $N_{inML}$ | Mixed layer nitrate input (($\times 10^{-7})\ mmol\ N\ m^{-2}\ s^{-1}$) | 4 [9, 10] |
| $\gamma$ | Nitrate content of phytoplankton ($mmol\ N$ per mg Chl) | 1/1.59 [11, 12] |
| $\lambda$ | Proportion of integrated chlorophyll below surface mixed layer | 0.9 |
| $z_s$ | Depth of surface mixed layer ($m$) | 30 (10-80) [13, 14] |
| $z_b$ | Bottom boundary of model domain ($m$) | 200 |
| $\frac{dN}{dz}\vert_{z=z_b}$ | Nitrate gradient at the bottom boundary of model domain ($mmol\ N\ m^{-4}$) | 0.2 (0-0.2) [15, 16, 17] |

Superscripts refer to the references that provide the source for the parameter value and the citations are as follows: [1]http://oceandata.sci.gsfc.nasa.gov/SeaWiFS/Mapped/Annual/9km/; [2]Wu and Gao, 2011; [3]Lee Chen et al., 2005; [4]Raven and Richardson, 1986; [5]Lu et al., 2010; [6]Bienfang and Harrison, 1984; [7]Liu et al., 2007; [8]Eppley et al., 1969; [9]Kim et al., 2014; [10]Duce et al., 2008; [11]Cloern et al., 1995; [12]Oschlies, 2001; [13]Wong et al., 2002; [14]Tseng et al., 2005; [15]Chen et al., 2006; [16]Our observations; [17]Li et al., 2015.

Table 2 Estimated results and observed values at SEATS station

| Variables | Estimated results | Observations |
|---|---|---|
| Nitracline depth (m) | 86 | 20-90[1, 2, 3] |
| Nitracline steepness (*mmol N m*$^{-4}$) | 0.21 | 0.30-0.50[3] |
| Depth of SCML (m) | 70 | 10-75[4, 5, 6] |
| Intensity of SCML (mg m$^{-3}$) | 0.47 | 0.40-0.90[4, 5, 6] |
| Thickness of SCML (m) | 24 | 10-55 [4, 5, 6] |

Superscripts refer to the references that provide the source for the parameter value and the citations are as follows: [1]Tseng et al., 2005; [2]Wong et al., 2007; [3]Our observations; [4]Chen et al., 2006; [5]Liu et al., 2002; [6]Liu et al., 2007.

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
