# Peer review of "Analytical solution of nitracline with the evolution of subsurface"

_Biogeosciences, 2016_

## Referee Comment (RC1) · A. W. Omta (Referee) · 21 Sep 2016

The manuscript is an analytical study of the relationship between the vertical distributions of phytoplankton and nutrients. An earlier paper by Gong et al. (2015) investigated the impact of light intensity, vertical diffusion, and the phytoplankton sinking velocity on the depth and width of the subsurface biomass maximum. Now, Gong et al. expand upon this earlier work with a careful study of what may determine the nutricline depth. The overall setup is good and there is a logical progression in the development of the text. Although analytical studies such as this one tend to be somewhat difficult to read, in my opinion they ought to have a much more prominent place in the field than they currently have, because they can provide much deeper insights than either

(forward) numerical simulations or (inverse) parameter/state estimations. Having said all this, I think that at two points in the study, some further analysis is warranted before publication:

1) The authors admit that the assumption that the chlorophyll distribution represents the phytoplankton biomass distribution "is a significant simplification. In fact, phytoplankton increases inter-cellular pigment concentration when light level decreases (Cullen, 1982; Fennel and Boss, 2003; Cullen, 2015)." (p. 6, l. 129-131) Now, there happen to be fairly precise mathematical descriptions of this effect, e.g., Cloern et al. (1995). Thus, the authors ought to be able to investigate how and to which extent photoacclimation would impact their predictions regarding the relationship between the subsurface chlorophyll maximum and the nutricline depth.

2) An unexpected prediction is the possible existence of nitrate minima below the surface mixed layer. According to the authors, these features disappear "if the subsurface vertical diffusion is too weak or the surface mixed layer is deeper than depth $z_{n1}$. The possible mechanism deserves to be explored." (p. 24/25, l. 606-608) I think I may understand the origin of these remarkable features. Consider a situation without phytoplankton sinking and with full recycling of dead phytoplankton (w=0, alpha=1). In that case, the nitrate distribution is simply the inverse of the phytoplankton distribution: if P has a maximum, then N has a minimum. When the sinking speed w increases and/or the recycling alpha decreases, a background vertical N gradient develops which makes the N minimum shallower, until it has disappeared. Essentially, the N minima are then the result of the phytoplankton eating holes in the N distributions. All this is illustrated in the attached figure. In my view, it would be very interesting, if the authors would investigate this hypothesis by varying the sinking velocity and the recycling coefficient, starting from w=0, alpha=1.

References

Cloern, J.E., C. Grenz, and L. Vidergar-Lucas, Limnology & Oceanography 40: 1313-

1321 (1995)

Cullen, J.J., Canadian Journal of Fish and Aquatic Sciences 39: 791-803 (1982)

Cullen, J.J., Annual Review of Marine Science 7: 207-239 (2015)

Fennel, K., and E. Boss, Limnology & Oceanography 48: 1521-1534 (2003)

Gong, X., J. Shi, H.W. Gao, and X.H. Yao, Biogeosciences 12: 905-919 (2015)

[Figure]

**Fig. 1.** The N minimum becomes shallower and eventually disappears as the sinking speed w increases.

---

## Referee Comment (RC2) · Anonymous Referee #2 · 30 Nov 2016

**1   General comments**

The study of Xiang Gong and coauthors is concerned with the existence and characteristics of a nitracline in the presence of subsurface chlorophyll maxima (SCM). The authors derive analytical solutions that describe possible steady state results of a one-dimensional vertical model of nutrients (dissolved inorganic nitrogen) and phytoplankton biomass. Analytical steady state solutions are nicely derived for stratified conditions, with some weak mixing below a shallow upper mixed layer. A piecewise function is introduced as an approximation of the vertical distribution of phytoplankton biomass. This elegant approach was described and applied in an earlier study by

X. Gong, J. Shi, H. W. Gao, and X. H. Yao, published 2015 in Biogeosciences, 12, 905-919. The authors take various different perspectives on the steady state solution. One of their main conclusions is that nitrate consumption by the phytoplankton has to be replenished by an upward flux of nitrate, which is interpreted as the major contribution to new primary production.

It is still fascinating to realise how much can be learned from analytical solutions of a model. X. Gong and his coauthore derive and explore steady state model solutions, elucidating interrelations between the characteristics of the nitracline and SCM. The stepwise derivation of particular solutions is generally good, but some readers may eventually lose track of all initial/original model assumptions. While reading about half of their study it became increasingly difficult to understand the actual meaning of the derived solutions, albeit mathematical steps were reproducible in most cases. For example, after the introduction of the depth of maximum growth ($z_0$), many statements are made and conclusions are drawn that may lead readers astray. The authors tend to interpret their analytical solutions to be indicative for true conditions. But the solutions only reflect steady state conditions of model results. Furthermore, the authors give the impression that their analytical solutions are straightforward and can be used to make inference about nitracline features, once $z_m$, h, and $\sigma$ have been derived from observed profiles of chlorophyll *a* (Chl *a*) concentration. To do so would be inappropriate, which should be explicitly stated in the study. It is a conceptual problem that has to be reasonably addressed by the authors. Some major revision of the manuscript is therefore needed before the study can be recommended for publication in Biogeosciences.

The analytical solutions presented are, apart from Equation (18) (see specific comments), correct. However, the author's should stress that the analytical solutions are valid only for estimates of $z_m$, h, and $\sigma$ that are consistent with the model's

numerical steady state solution. The numerical steady state solution in turn depends on the forcing, boundary conditions and on the combination of parameter values. The approximations of $z_m$, h, and $\sigma$ are entirely conditioned by the model results and thus also depend on the combination of model parameter values. To combine the analytical steady state solutions with observed $z_m$, h, and $\sigma$ (as derived from vertical profiles of chlorophyll *a* concentration) is only meaningful after model calibration (identifying a model solution that is in some agreement with the observed $z_m$, h, and $\sigma$). A calibration requires the numerical model to be run in the first place. In other words, the equations, e.g. for the depth of the nitracline ($z_n$), are valid only for $z_m$, h, and $\sigma$ that remain dynamically consistent with the imposed model. Otherwise, the derived equations are not applicable.

Another concern is, although already addressed/discussed by the authors, the neglect of photoacclimation dynamics. The process of photoacclimation is essential for those systems (with stratified conditions) the authors focus on, and such a model approach would be better suited to make inference about the basic interrelations between a nitracline and a SCM. A possibility would be to include some additional parameterisation that could yield variable $\gamma$, which can be derived from e.g. Cloern et al. (1995, L&O, 40(7), 1313-1321). When resorting to a parameterisation of Cloern et al. (e.g. their Eq. 15), some care has to be taken only with respect to the temporal integral of daily irradiance that is averaged over the upper mixed layer in their study. A certainly more realistic model would be one with equations that explicitly resolve variations of the Chl *a*-to-carbon and nitrogen-to-carbon ratio of the algae. An interesting aspect would be to see whether the "symmetric", piecewise Gaussian function would still be useful to approximate profiles of simulated Chl *a*, even if still applicable to fit phytoplankton nitrogen biomass. The authors only discuss possible shifts in depth (location) of the SCM. They do not consider skewed profiles of Chl *a*, with a sharp SCM, as can be seen in many Chl *a* observational profiles.

**2 Specific comments**

**Abstract**
lines 26-27: "..., we derive analytical solutions for the system of phytoplankton and nutrient."
The authors derive analytical solutions of a specified model. The model is well suited to explain basic dependencies between a nitracline and a deep chlorophyll *a* maximum.
lines 31-34: "The inverse proportional relationship..., suggesting that the light level at the nitracline can be used as an indicator for integrated new primary production."
It is not clear whether the model approach is appropriate to clearly distinguish between regenerated and new production. The dynamical model equations only resolve some instant remineralisation, with a direct mass flux from the phytoplankton back to the nutrient pool.

**1 Introduction**
The introduction is nice. It is well written and informative.
line 112: "... was used to fit vertical chlorophyll profiles."
Here the authors should clarify that the Gaussian function is used as a fit to the steady state solution of the model.

**2 Definition and models**
pages 5 - 9: The model is nicely described and sufficient details are provided. I would suggest to introduce $\lambda$ not here but where it is needed (on page 18).
page 9, lines 235 - 237: "We use the biologically reasonable parameter values given in Table 1 to represent the system at station SEATS..."
Thus, a specific (calibrated) model solution is considered as an example.

pages 10 - 11: **Definition of the nitracline**

The text is well written. The concept described in the final paragraph (lines 270 - 280) is clear. However, it is still confusing because simulated as well as observed profiles of N yield $\frac{d^2 N}{dz^2} \approx 0$ (or $\frac{dN}{dz} \approx$ constant) over some distinct depth range, e.g. as depicted in Fig. (2).

The described balance between uptake and recycling only works for this particular kind of model approach. The authors may add "According to our model approach (Eq. 2) the depth where $\frac{d^2 N}{dz^2} = 0$ represents a balance between the growth rate and the phytoplankton loss rate."

**3 Results**

page 12: You may add here the depth range that is considered ($z_s < z < z_b$).

line 112: "... the fitted function of chlorophyll..."

Suggestion: "... the fitted, depth dependend function of chlorophyll ($\gamma P(z)$). This reminds the reader that $P$ actually includes an exponential in Eq. (8).

page 13: The minus sign (- $K_{\nu 2}/\sigma^4$) is confusing.

line 313: do the authors mean... "... (values from 8.64 to 7.78 $\cdot$ $10^{-9}$ m$^{-2}$ s$^{-1}$)..."?

Equation (12): for non-zero $w$

**Depth of the nitracline**

page 15, lines 353 - 364: This derivation only works when Blackman's law of limiting factors (light and nutrient limitation) is applied. Hence, it is a particular model assumption. The maximum rate discussed here first of all represents a net primary production term. Only in the context of this particular model version it is also interpreted as new primary production. The sentence "It follows that the light level at the nitracline is an indicator of integrated NPP in the water column." is a strong statement. This finding strongly depends on the underlying model equations. It would be good to see different steady state solutions of the model while varying values of $\epsilon$ and $\alpha$ (e.g. increasing $\epsilon$ while decreasing $\gamma$ and vice versa). This way the authors may substantiate their

conclusion.

Equation (18): The inclusion of $\gamma$ in the last term is incorrect. The parameter $\gamma$ can be removed. This is because $K_c$ is normalised to nitrogen biomass and not to Chl *a*.

page 16, lines 377 - 380: "Equation (18) also indicates that both a higher recycling rate ($\alpha$) of dead phytoplankton and a larger loss rate ($\epsilon$)lead to a shallower nitracline, while the enhanced maximum growth rate of the phytoplankton ($\mu_m$) moves the nitracline depth down."

It would be good to see this conclusion consolidated by some model results. This way the authors can also demonstrate the predictive power of applying Eq. (18). The parameters could be varied just as discussed by the authors and it would be interesting to see how well an updated $z_n$ (based on the model runs with the parameter values varied) matches the predicted $z_n$ of Eq. (18) (based on the previous model results, e.g. of P).

page 17, lines 416 - 419: "Our results indicate... self-shading negatively influences depth and thickness of the SCML,..."

This is comprehensible.

**4 Discussion**

**In presence of surface nutrient input**

page 22, lines 527 - 535: This is certainly the case for the model assumption of an instant remineralisation of organic matter that originates directly from the phytoplankton. Must this (the need to include a surface nutrient source) also be expected for a model approach where dissolved organic matter (DOM) and detritus are explicitly resolved?

**Vertical profiles of nitrate gradients**

page 24, line 605: In Fig. (4) the profile of $\frac{dN}{dz} \cdot 20$ does not correspond with the shown profile of N. The N profile clearly indicates a constant $\frac{dN}{dz}$ (of approximately 0.38 mmol

N m$^{-4}$ $\longrightarrow$ 7.6 mmol N m$^{-4}$ = $\frac{dN}{dz}$ · 20) in the depth range of 50 -70 m. The shown $\frac{dN}{dz}$ · 20 does not reveal this feature. The authors need to clarify this.

**Limitation and application**
page 24, lines 610- 636: As important as the model assumptions for the sinking and remineralisation of particulate organic matter is photoacclimation. The authors should consider to include one or two figures with profiles of Chl *a* concentrations with typical but different shapes of the SCM.

page 26, lines 648 - 657: I used the parameter values of Table (1) and the values for $z_m$, $h$, and $\sigma$ from Table (2) to calculate the corresponding $z_n$ and $\frac{dN}{dz}$ (Eqs. 11 and 14). I obtain $z_n$=70 m and $\frac{dN}{dz}$ = 0.025 mmol N m$^{-4}$. In Table the solutions are $z_n$=79 m and $\frac{dN}{dz}$ = 0.24 mmol N m$^{-4}$. I cross-checked my equations and all values and have not found any explanation for this discrepancy. I thought that all values presented are consistent with the imposed model dynamics and thus valid for any of the analytical steady state solutions presented.

**Summary**
pages 26 - 27: The authors may here stress that the important findings are conditioned by the model equations imposed. The interpretation of NPP is not straightforward and becomes particularly difficult to specify under steady state conditions of a weakly mixed water column. The authors construct NPP from the model equations that rely in Blackman's law of limiting factor for the growth rate. I suggest to the authors to refine their statements, clarifying their findings are based on the assumption that a prominent instant recycling process exists.

---

## Author Response (AR1)

[revised manuscript text omitted]

**List of what we changed in the revised version of manuscript:**

1005

| Original | Revised |
|---|---|
| Page 1, line 27-28 | **Line 27-29**: Change "we derive analytical solutions for the system of phytoplankton and nutrient." to "we derive analytical solutions of a specified nutrient-phytoplankton model. The model is well suited to explain basic dependencies between a nitracline and a SCML." |
| Page 2, line 33-34 | Delete the sentence "suggesting that the light level at the nitracline can be used as an indicator for integrated new primary production." |
| Page 5, line 110-112 | **Line 110-113**: Change "Accordingly, a piecewise function comprising a constant value within the surface mixed layer and a Gaussian function below this layer was used to fit vertical chlorophyll profiles." To "Accordingly, a piecewise function comprising a constant value within the surface mixed layer and a Gaussian function below this layer was used as a fit to the steady state vertical chlorophyll profiles simulated by the nutrient-phytoplankton model." |
| Page 10, after line 237 | **Line 232-233**: Add a sentence "Thus, the specific (calibrated) model solution is considered as an example to obtain the analytic solutions of nitracline." |
| Page 11, line 273-275 | **Line 269-271**: Rewrite the definition of nitracline depth "In this study, we adopt the location of the maximum nitrate gradient ($\max(dN/dz)$) in the euphotic zone as the nitracline depth ($z_n$), which can be expressed by $\left.\dfrac{d^2N}{dz^2}\right|_{z_n} = 0$ and $\left.\dfrac{d^3N}{dz^3}\right|_{z_n} < 0$." |

| | |
|---|---|
| Page 11, line 277-278 | **Line 273-275**: Change the sentence "Thus, at the nitracline depth the balance between uptake and recycling terms can be derived: $\mu m\min(f(I),\ g(N))=\alpha\varepsilon$." To "Thus, according to our model approach (Eq. 2) the nitracline depth where $\dfrac{d^2N}{dz^2}=0$ represents a balance between the nutrient uptake and the recycling of phytoplankton loss, i.e., $\mu_m\min(f(I),\ g(N))=\alpha\varepsilon$." |
| Page 12, line 286-288 | **Line 283-286**: Add the explanation of ISUS data processing method "The sampling frequency was set at 5 Hz and the raw data were thus smoothed with a 25-point moving average in the surface mixed layer, a 5-point moving average in the SCML, and a 15-point moving average below the SCML. The data were then interpolated by a cubic spline function." |
| Page 12, line 296-297 | Spell out chlorophyll concentration P is a fitted and depth dependent variable, **Line 295-296**: "the fitted, depth dependent function of chlorophyll ($P(z)$, Eq. 6)". |
| Page 12, line 299 and 301 | **Line 298 and 300**: Add the domain of Eqs. (8) and (9), i.e., " $z_s < z < z_b$ ". |
| Page 13, line 310 | **Line 309**: Delete the minus sign in $-K_{v2}/\sigma^4$. |
| Page 14, line 335 and 339 | **Line 200**: Correct Eqs. (13) and (14) as $\left.\dfrac{dN}{dz}\right|_{z_n} = \left.\dfrac{dN}{dz}\right|_{z_b} + \left(\dfrac{z_n-z_m}{\sigma^2}+\dfrac{w}{K_{v2}}\right)\gamma P\big|_{z_n} - \dfrac{(1-\alpha)\varepsilon\gamma}{K_{v2}}\int_{z_n}^{z_b}Pdz$ and $\left.\dfrac{dN}{dz}\right|_{z_n} = \left.\dfrac{dN}{dz}\right|_{z_b} + \left(\sqrt{\dfrac{w^2}{4K_{v2}^2}+\dfrac{(1-\alpha)\varepsilon}{K_{v2}}+\dfrac{1}{\sigma^2}}+\dfrac{w}{2K_{v2}}\right)\gamma P\big|_{z_n} - \dfrac{(1-\alpha)\varepsilon\gamma}{K_{v2}}\int_{z_n}^{z_b}Pdz$ |
| Page 15, line 359 | **Line 358-359**: Spell out the condition by Eq. (16) "Note that the derivation of Eq. (16) only works when light and nutrient limitation (Blackman's law of limiting factors) is applied." |

[revised manuscript text omitted]

| | |
|---|---|
| | chlorophyll concentration in the whole water column and with the maximum rate of NPP, acting as the indicator of integrated NPP. The NPP is constructed from the model equations that rely in Blackman's law of limiting factor for the growth rate. These findings are based on the assumption that a prominent instant recycling process exists." |
| Acknowledgements | **Line 729-732**: Add many thanks to the two referees and other three friends "We are very grateful to A. W. Omta and another anonymous reviewer for their constructive and helpful suggestions. We also would like to thank Xiaohuan Liu, Yang Yu, and Xiaokun Ding for valuable advice and programming assistance." |
| References | Add 3 references in **line 826-827, 954-956, 978-979.** |
| Figures | Redraw the profiles of ISUS Nitrate and Nitrate gradient by using new data processing method in Fig. 4. |
| Tables | Correct the wrong typo of $\gamma$ in Table 1.
Correct the estimated results in Table 2. |
*The manuscript is an analytical study of the relationship between the vertical distributions of phytoplankton and nutrients. An earlier paper by Gong et al. (2015) investigated the impact of light intensity, vertical diffusion, and the phytoplankton sinking velocity on the depth and width of the subsurface biomass maximum. Now, Gong et al. expand upon this earlier work with a careful study of what may determine the nutricline depth. The overall setup is good and there is a logical progression in the development of the text. Although analytical studies such as this one tend to be somewhat difficult to read, in my opinion they ought to have a much more prominent place in the field than they currently have, because they can provide much deeper insights than either (forward) numerical simulations or (inverse) parameter/state estimations. Having said all this, I think that at two points in the study, some further analysis is warranted before publication:*

**Response:** We are very grateful for the helpful comments and we have revised our manuscript accordingly.

*1) The authors admit that the assumption that the chlorophyll distribution represents the phytoplankton biomass distribution "is a significant simplification. In fact, phytoplankton increases inter-cellular pigment concentration when light level decreases (Cullen, 1982; Fennel and Boss, 2003; Cullen, 2015)." (p. 6, l. 129-131) Now, there happen to be fairly precise mathematical descriptions of this effect, e.g., Cloern et al. (1995). Thus, the authors ought to be able to investigate how and to which extent photoacclimation would impact their predictions regarding the relationship between the subsurface chlorophyll maximum and the nutricline depth.*

**Response:** Agree. We parameterize Chl: C using Eq. 15 of *Cloern et al.* Then let R= Chl: C, the nitrogen content of phytoplankton $\gamma$ will be written as $\gamma= 1/(6.625*12*R)$, corresponding to a C:N ratio of 6.625 and a carbon atomic mass of 12. The detailed results have been added in Section 4.5 to illustrate how and to which extent photoacclimation influence the relationships between a nitracline and a SCM.

*2) An unexpected prediction is the possible existence of nitrate minima below the surface mixed layer. According to the authors, these features disappear "if the subsurface vertical diffusion is too weak or the surface mixed layer is deeper than depth zn1. The possible mechanism deserves to be explored." (p. 24/25, l. 606-608). I*

*think I may understand the origin of these remarkable features. Consider a situation without phytoplankton sinking and with full recycling of dead phytoplankton (w=0, alpha=1). In that case, the nitrate distribution is simply the inverse of the phytoplankton distribution: if P has a maximum, then N has a minimum. When the sinking speed w increases and/or the recycling alpha decreases, a background vertical N gradient develops which makes the N minimum shallower, until it has disappeared. Essentially, the N minima are then the result of the phytoplankton eating holes in the N distributions. All this is illustrated in the attached figure. In my view, it would be very interesting, if the authors would investigate this hypothesis by varying the sinking velocity and the recycling coefficient, starting from w=0, alpha=1.*

**Response:** Many thanks for this suggestion. Following this idea, we adopted numerical simulation to examine this hypothesis by varying the sinking velocity and the recycling coefficient. The results have been added in the revision.
*The study of Xiang Gong and coauthors is concerned with the existence and characteristics of a nitracline in the presence of subsurface chlorophyll maxima (SCM). The authors derive analytical solutions that describe possible steady state results of a one-dimensional vertical model of nutrients (dissolved inorganic nitrogen) and phytoplankton biomass. Analytical steady state solutions are nicely derived for stratified conditions, with some weak mixing below a shallow upper mixed layer. A piecewise function is introduced as an approximation of the vertical distribution of phytoplankton biomass. This elegant approach was described and applied in an earlier study by X. Gong, J. Shi, H. W. Gao, and X. H. Yao, published 2015 in Biogeosciences, 12, 905-919. The authors take various different perspectives on the steady state solution. One of their main conclusions is that nitrate consumption by the phytoplankton has to be replenished by an upward flux of nitrate, which is interpreted as the major contribution to new primary production.*

*It is still fascinating to realise how much can be learned from analytical solutions of a model. X. Gong and his coauthors derive and explore steady state model solutions, elucidating interrelations between the characteristics of the nitracline and SCM. The stepwise derivation of particular solutions is generally good, but some readers may eventually lose track of all initial/original model assumptions. While reading about half of their study it became increasingly difficult to understand the actual meaning of the derived solutions, albeit mathematical steps were reproducible in most cases. For example, after the introduction of the depth of maximum growth ($z_0$), many statements are made and conclusions are drawn that may lead readers astray. The authors tend to interpret their analytical solutions to be indicative for true conditions. But the solutions only reflect steady state conditions of model results. Furthermore, the authors give the impression that their analytical solutions are straightforward and can be used to make inference about nitracline features, once $z_m$, $h$, and $\sigma$ have been derived from observed profiles of chlorophyll a (Chl a) concentration. To do so would be inappropriate, which should be explicitly stated in the study. It is a conceptual problem that has to be reasonably addressed by the authors. Some major revision of the manuscript is therefore needed before the study can be recommended for publication in Biogeosciences.*

**Response:** Many thanks for the helpful suggestions and comments. We try best to revise and make physical meanings more obvious with those derived solutions. We also tone down the statements and conclusions to avoid any misleading. For example, we moved the statements drawn from Equation (17) (line 361-364) to the Discussion. The challenge and uncertainty have been included when we present those implications. Please see the revision.

*The analytical solutions presented are, apart from Equation (18) (see specific comments), correct. However, the author's should stress that the analytical solutions are valid only for estimates of $z_m$, h, and σ that are consistent with the model's numerical steady state solution. The numerical steady state solution in turn depends on the forcing, boundary conditions and on the combination of parameter values. The approximations of $z_m$, h, and σ are entirely conditioned by the model results and thus also depend on the combination of model parameter values. To combine the analytical steady state solutions with observed $z_m$, h, and σ (as derived from vertical profiles of chlorophyll a concentration) is only meaningful after model calibration (identifying a model solution that is in some agreement with the observed $z_m$, h, and σ). A calibration requires the numerical model to be run in the first place. In other words, the equations, e.g. for the depth of the nitracline ($z_n$), are valid only for $z_m$, h, and σ that remain dynamically consistent with the imposed model. Otherwise, the derived equations are not applicable.*

**Response:** Agree. We state that the analytical steady state solutions of nitracline are applicable only for estimates of $z_m$, h, and σ that are consistent with the model's numerical steady state solution. The challenge and uncertainty have been included when we present those implications.

*Another concern is, although already addressed/discussed by the authors, the neglect of photoacclimation dynamics. The process of photoacclimation is essential for those systems (with stratified conditions) the authors focus on, and such a model approach would be better suited to make inference about the basic interrelations between a nitracline and a SCM. A possibility would be to include some additional parameterization that could yield variable γ, which can be derived from e.g. Cloern et al. (1995, L&O, 40(7), 1313-1321). When resorting to a parameterisation of Cloern et al. (e.g. their Eq. 15), some care has to be taken only with respect to the temporal integral of daily irradiance that is averaged over the upper mixed layer in their study. A certainly more realistic model would be one with equations that explicitly resolve variations of the Chl a-to-carbon and nitrogen-to-carbon ratio of the algae. An interesting aspect would be to see whether the "symmetric", piecewise Gaussian function would still be useful to approximate profiles of simulated Chl a, even if still*

*applicable to fit phytoplankton nitrogen biomass. The authors only discuss possible*
*shifts in depth (location) of the SCM. They do not consider skewed profiles of Chl a,*
*with a sharp SCM, as can be seen in many Chl a observational profiles.*

**Response:** In the revision, we parameterize Chl: C using Eq. 15 of *Cloern et al.*
(1995). Then let R= Chl: C, the nitrogen content of phytoplankton $\gamma$ will be written as
$\gamma = 1/(6.625*12*R)$, corresponding to a C:N ratio of 6.625 and a carbon atomic mass
of 12. The detailed results will be added in Section 4.5 to illustrate how and to which
extent photoacclimation influence the relationships between a nitracline and a SCM.
The limitation, i.e., the skewed profiles of Chl a with a sharp SCM was not considered,
have been added in Section 4.5.

**2 Specific comments**

**Abstract**
*lines 26-27: "..., we derive analytical solutions for the system of phytoplankton and*
*nutrient."*
*The authors derive analytical solutions of a specified model. The model is well suited*
*to explain basic dependencies between a nitracline and a deep chlorophyll a*
*maximum.*

**Response:** Agree. This sentence have been rewritten, i.e., we derive analytical
solutions of a specified nutrient-phytoplankton model. The model is well suited to
explain basic dependencies between a nitracline and a SCML.

*lines 31-34: "The inverse proportional relationship..., suggesting that the light level*
*at the nitracline can be used as an indicator for integrated new primary production."*
*It is not clear whether the model approach is appropriate to clearly distinguish*
*between regenerated and new production. The dynamical model equations only*
*resolve some instant remineralisation, with a direct mass flux from the phytoplankton*
*back to the nutrient pool.*

**Response:** Agree. We delete this sentence and modify the related results in the text
accordingly.

**1 Introduction**
*The introduction is nice. It is well written and informative.*
*line 112: "... was used to fit vertical chlorophyll profiles."*
*Here the authors should clarify that the Gaussian function is used as a fit to the steady*
*state solution of the model.*

**Response:** Agree. We spell out that the Gaussian function is used as a fit to the steady
state solution of the model. In reality, the Gaussian function is also applicable for

many profiles of Chl a in stratified waters, especially open ocean.

110 **2 Definition and models**

*pages 5 - 9: The model is nicely described and sufficient details are provided. I would suggest to introduce λ not here but where it is needed (on page 18).*

**Response:** Agree. We move the introduction of $\lambda$ to page 18.

*page 9, lines 235 - 237: "We use the biologically reasonable parameter values given*
115 *in Table 1 to represent the system at station SEATS..."*
*Thus, a specific (calibrated) model solution is considered as an example.*

**Response:** Agree. We add this sentence in the revision.

pages 10 - 11: **Definition of the nitracline**

*The text is well written. The concept described in the final paragraph (lines 270 - 280)*
120 *is clear. However, it is still confusing because simulated as well as observed profiles of N yield $\frac{d^2N}{dz^2} \approx 0$ (or $\frac{dN}{dz} \approx$ constant) over some distinct depth range, e.g. as depicted in Fig. (2).*

**Response:** Many thanks for noticing this issue. The depth of nitracline in our study was defined as the location of maximum nitrate gradient in the euphotic zone, which
125 can be expressed by $\frac{d^2N}{dz^2} = 0$ and $\frac{d^3N}{dz^3} < 0$, not implying $\frac{dN}{dz} =$ constant. The equality $\frac{d^2N}{dz^2} = 0$ means $\frac{dN}{dz} \approx$ constant only when $\frac{d^2N}{dz^2} \equiv 0$ in the domain. Thus, to determine the depth of nitracline from simulated as well as observed profiles, we have to plot the profile of nitrate gradient and find the maximum nitrate gradient ($\frac{dN}{dz}|_{\max}$). We have rewritten this sentence in the revision.

130 *The described balance between uptake and recycling only works for this particular kind of model approach. The authors may add "According to our model approach (Eq. 2) the depth where $\frac{d^2N}{dz^2}=0$ represents a balance between the growth rate and the phytoplankton loss rate."*

**Response:** Agree. This has been added in the revision.

135 **3 Results**

*page 12: You may add here the depth range that is considered ($z_s < z < z_b$).*

**Response:** Agree. The depth range $z_s < z < z_b$ has been added in Eq. (8).

*line 112: "... the fitted function of chlorophyll..."*

*Suggestion: "... the fitted, depth dependend function of chlorophyll (γP(z)). This reminds the reader that P actually includes an exponential in Eq. (8).*

**Response:** Agree. This sentence has been revised as "... the fitted, depth dependent function of chlorophyll (P(z))...".

*page 13: The minus sign (- $K_{v2}/\sigma^4$) is confusing.*

**Response:** We have delete the minus sigh in the revision.

*line 313: do the authors mean... "... (values from 8.64 to $7.78*10^{-9}$ $m^{-2}$ $s^{-1}$)..."?*

**Response:** Because $K_{v2}$ is $1\text{-}9*10^{-9}$ $m^2$ $s^{-1}$, $\sigma$ is from several meters to tens of meters, thus the ratio of $K_{v2}$ to $\sigma^4$ is from $8.64*10^{-9}$ to $7.78$ $\cdot m^{-2}$ $s^{-1}$. This will be clarified in the revision.

*Equation (12): for non-zero w.*

**Response:** We add the condition of non-zero $w$ for Eq. (12) in the revision.

**Depth of the nitracline**

*page 15, lines 353 - 364: This derivation only works when Blackman's law of limiting factors (light and nutrient limitation) is applied. Hence, it is a particular model assumption. The maximum rate discussed here first of all represents a net primary production term. Only in the context of this particular model version it is also interpreted as new primary production. The sentence "It follows that the light level at the nitracline is an indicator of integrated NPP in the water column." is a strong statement. This finding strongly depends on the underlying model equations. It would be good to see different steady state solutions of the model while varying values of $\varepsilon$ and α (e.g. increasing $\varepsilon$ while decreasing α and vice versa). This way the authors may substantiate their conclusion.*

**Response:** We spell out that the derivation only works when Blackman's law of limiting factors (light and nutrient limitation) is applied in the revision. We will also examine the simulated results by varying values of $\varepsilon$ and α, please see the revision.

*Equation (18): The inclusion of γ in the last term is incorrect. The parameter γ can be removed. This is because $K_c$ is normalised to nitrogen biomass and not to Chl a.*

**Response:** Sorry for the typo, we have removed "γ".

*page 16, lines 377 - 380: "Equation (18) also indicates that both a higher recycling rate (α) of dead phytoplankton and a larger loss rate ($\varepsilon$) lead to a shallower nitracline, while the enhanced maximum growth rate of the phytoplankton ($\mu_m$) moves*

*the nitracline depth down."*

*It would be good to see this conclusion consolidated by some model results. This way the authors can also demonstrate the predictive power of applying Eq. (18). The parameters could be varied just as discussed by the authors and it would be interesting to see how well an updated $z_n$ (based on the model runs with the parameter values varied) matches the predicted $z_n$ of Eq. (18) (based on the previous model results, e.g. of P).*

**Response:** Thank you for this suggestion. We run the N-P model to examine how well the modelled $z_n$ matches the predicted value and the results have been added in the revision.

*page 17, lines 416 - 419: "Our results indicate... self-shading negatively influences depth and thickness of the SCML,..."*

*This is comprehensible.*

**Response:** We rewrite this sentence.

**4 Discussion**

**In presence of surface nutrient input**

*page 22, lines 527 - 535: This is certainly the case for the model assumption of an instant remineralisation of organic matter that originates directly from the phytoplankton.*

*Must this (the need to include a surface nutrient source) also be expected for a model approach where dissolved organic matter (DOM) and detritus are explicitly resolved?*

**Response:** After examining the N-P-D model given by Beckmann and Hense (2007), we found that the surface nutrient input is not necessary for a model approach where dissolved organic matter (DOM) and detritus are explicitly resolved. In the revision, we will spell out this assumption.

**Vertical profiles of nitrate gradients**

*page 24, line 605: In Fig. (4) the profile of $\frac{dN}{dz} \cdot 20$ does not correspond with the shown profile of N. The N profile clearly indicates a constant $\frac{dN}{dz}$ (of approximately 0.38 mmol N $m^{-4} \rightarrow$ 7.6 mmol N $m^{-4} = \frac{dN}{dz} \cdot 20$) in the depth range of 50 -70 m. The shown $\frac{dN}{dz} \cdot 20$ does not reveal this feature. The authors need to clarify this.*

**Response:** Thank your comments. We redraw this figure by using new data

processing method in the revision.

**Limitation and application**

*page 24, lines 610- 636: As important as the model assumptions for the sinking and remineralisation of particulate organic matter is photoacclimation. The authors should consider to include one or two figures with profiles of Chl a concentrations with typical but different shapes of the SCM.*

**Response:** In the revision, we spell out the limitation of photoacclimation in this Section and revise the manuscript accordingly.

*page 26, lines 648 - 657: I used the parameter values of Table (1) and the values for $z_m$, h, and σ from Table (2) to calculate the corresponding $z_n$ and $\frac{dN}{dz}$ (Eqs. 11 and 14). I obtain $z_n=70$ m and $\frac{dN}{dz} = 0.025$ mmol N $m^{-4}$. In Table the solutions are $z_n=79$ m and $\frac{dN}{dz} = 0.24$ mmol N $m^{-4}$. I cross-checked my equations and all values and have not found any explanation for this discrepancy. I thought that all values presented are consistent with the imposed model dynamics and thus valid for any of the analytical steady state solutions presented.*

**Response:** Sorry for the typo. We found that the value of $\gamma$ should be 1/1.59, not 1.59. We will recalculate the values. Please see the revision.

**Summary**

*pages 26 - 27: The authors may here stress that the important findings are conditioned by the model equations imposed. The interpretation of NPP is not straightforward and becomes particularly difficult to specify under steady state conditions of a weakly mixed water column. The authors construct NPP from the model equations that rely in Blackman's law of limiting factor for the growth rate. I suggest to the authors to refine their statements, clarifying their findings are based on the assumption that a prominent instant recycling process exists.*

**Response:** Thank you for the helpful suggestions. We rewrite the summary by stressing the assumptions of our model results, especially the statements about NPP.